# LEARNING WHAT TO SAY AND HOW PRECISELY: EFFICIENT COMMUNICATION VIA DIFFERENTIABLE DISCRETE COMMUNICATION LEARNING

## ABSTRACT

Effective communication in multi-agent reinforcement learning (MARL) is critical for success but constrained by bandwidth, yet past approaches have been limited to complex gating mechanisms that only decide *whether* to communicate, not *how precisely*. Learning to optimize message precision at the bit-level is fundamentally harder, as the required discretization step breaks gradient flow. We address this by generalizing Differentiable Discrete Communication Learning (DDCL), a framework for end-to-end optimization of discrete messages. Our primary contribution is an extension of DDCL to support unbounded signals, transforming it into a universal, plug-and-play layer for any MARL architecture. We verify our approach with three key results. First, through a qualitative analysis in a controlled environment, we demonstrate *how* agents learn to dynamically modulate message precision according to the informational needs of the task. Second, we integrate our variant of DDCL into four state-of-the-art MARL algorithms, showing it reduces bandwidth by over an order of magnitude while matching or exceeding task performance. Finally, we provide direct evidence for the "Bitter Lesson" in MARL communication: a simple Transformer-based policy leveraging DDCL matches the performance of complex, specialized architectures, questioning the necessity of bespoke communication designs.

## 1 INTRODUCTION

Multi-agent reinforcement learning (MARL) has emerged as a powerful framework for training autonomous agents to solve complex tasks (Brown & Sandholm, 2019; Vinyals et al., 2011; Kober et al., 2013; Shalev-Shwartz et al., 2016; Zhou et al., 2020). In many scenarios, effective communication is crucial for high performance, particularly when individual observations are incomplete or decentralized coordination is required (Oliehoek & Amato, 2016b). For this reason, many MARL approaches focus on learning inter-agent communication strategies alongside behavioral policies (Foerster et al., 2016b; Peng et al., 2017; Sukhbaatar et al., 2016b; Freed et al., 2020b). However, real-world systems possess bandwidth constraints, making the efficient use of the communication network a primary concern. Our goal is to develop agents that maximize group performance while minimizing communication bandwidth.

Numerous approaches have been proposed for MARL with communication (MARL+Comms), often featuring bespoke neural architectures and training methods (Kim et al., 2019; Wang et al., 2020; Sukhbaatar et al., 2016a; Hu et al., 2024; Das et al., 2019; Niu et al., 2021; Liu et al., 2020; Singh et al., 2019). A major limitation of these approaches is the assumption of high-precision messages, typically 32-bit floating-point vectors (Han et al., 2023; Hu et al., 2020; Huang et al., 2016). While this enables training via gradient descent, it leads to excessive bandwidth usage. Some methods (Singh et al., 2019; Das et al., 2019; Liu et al., 2020) attempt to address this with hard attention mechanisms that gate communication, allowing agents to choose *whether* to send a message at a given timestep. While these approaches can reduce bandwidth, they optimize at the "frequency level". They do not allow agents to dynamically modulate the *precision* of their messages, only the frequency.

Differentiable Discrete Communication Learning (DDCL) (Freed et al., 2020c;a) is a recent approach that provides more fine-grained, bit-level control than prior approaches (Chaabouni et al., 2019; Havrylov & Titov, 2017). DDCL uses a stochastic quantization process to map real-valued vectors to variable-length bitstrings, where larger-magnitude vectors map to longer bitstrings. Crucially, the expected message length is a differentiable function of the message vector, allowing bandwidth to be minimized directly via gradient descent. This allows agents to modulate message precision based on their observations. However, prior work on DDCL is limited; it has only been demonstrated in simple tasks and relies on restrictive assumptions that communication vectors must be positive and bounded, constraining its expressivity.

In this work, we unlock the full potential of DDCL as a general-purpose tool for efficient MARL communication. We begin by generalizing the DDCL framework to support **unbounded, signed communication vectors**, removing the constraints on communication vector in prior DDCL work and enabling its integration into any MARL agent. We use this generalized framework to demonstrate not just *that* communication can be made more efficient, but *how* agents learn to allocate bandwidth.

Our work validates this generalized framework with a threefold contribution. First, we provide a qualitative analysis in an interpretable grid-world environment to demonstrate **how** our DDCL variant enables agents to learn to dynamically control message precision based on the informational needs of the task. Second, we demonstrate the **practical utility and generality** of our method by integrating it into four diverse MARL+Comms algorithms—IC3Net, TarMAC, GA-Comm, and MAGIC (Singh et al., 2019; Das et al., 2019; Liu et al., 2020; Niu et al., 2021)—and showing that across multiple benchmarks, it reduces communication bandwidth by over an order of magnitude while maintaining or improving task performance. Finally, we show that a simple, general-purpose encoder-only Transformer-based policy (Vaswani et al., 2023) empowered by DDCL can match or exceed the performance of complex, specialized communication architectures, suggesting that future research should prioritize scalable, general mechanisms over bespoke designs. This provides the first direct evidence for the "Bitter Lesson" in MARL communication.

## 2 PRELIMINARIES

### 2.1 PARTIALLY OBSERVABLE MARKOV GAME WITH COMMUNICATION

We model multi-agent problems as a Partially Observable Markov Game (POMG), defined by the tuple $\mathcal{G} = (\mathcal{S}, \{\mathcal{A}^i\}, \{R^i\}, P, \{O^i\}, \gamma)$. Here, $\mathcal{S}$ is the global state space and $\mathcal{A}^i$ is the action space for each agent $i$. At each timestep $t$, the environment is in state $s_t \in \mathcal{S}$. Each agent receives a private, partial observation $o_t^i \sim O^i(s_t)$ and selects an action $a_t^i$ based on its policy, $\pi^i$. The joint action of all agents induces a state transition $s_{t+1} \sim P(\cdot|s_t, a_t)$, and each agent receives a reward $r_t^i = R^i(s_t, a_t)$. We focus on the fully cooperative setting, where all agents share a common reward function, i.e., $R^i = R$ for all agents $i$.

In MARL with communication (MARL+Comms), this process is augmented. Before selecting an action, each agent $i$ can generate a message $m_t^i$ based on its private partial observation $o_t^i$. This message is then broadcast to other agents. The receiving agent $j$ incorporates the incoming message(s) $m_t^i$ into its own observation, forming an augmented observation that its policy uses to select its action $a_t^j$. The core challenge is learning a communication protocol and a behavioral policy simultaneously.

### 2.2 DIFFERENTIABLE DISCRETE COMMUNICATION LEARNING (DDCL)

Practical multi-agent systems require discrete communication protocols due to bandwidth-limited digital channels (Freed et al., 2020b; Chen et al., 2024; Tucker et al., 2022). Learning these protocols presents a dilemma. Treating messages as discrete actions naturally handles discrete channels, but learns inefficiently and often converges to inferior policies. Conversely, treating communication as a differentiable process allows for efficient end-to-end training via backpropagation but traditionally assumes continuous, real-valued messages, making it inapplicable to discrete channels without introducing biased gradient estimators.

Differentiable Discrete Communication Learning (DDCL) (Freed et al., 2020c;a) was introduced to resolve this conflict. Its core innovation is a stochastic encoding and decoding procedure that makes a discrete channel mathematically equivalent to an analog channel with additive noise, through which unbiased gradients can be backpropagated.

**The Reparameterization Mechanism.** The core mechanism for passing a message from a sender to a receiver begins with the sender's policy outputting a real-valued signal $z$, originally assumed to be a scalar in $[0, 1]$. The sender perturbs this signal with random noise $\epsilon \sim U(-\delta/2, +\delta/2)$, where $\delta$ is the quantization width, to produce a noisy signal $z' = z + \epsilon$. This noisy signal is then quantized into a discrete integer message $m = \lfloor \frac{(z') \pmod 1}{\delta} \rfloor$, which is sent over the non-differentiable channel. Upon receiving $m$, the receiver reconstructs the signal by computing the center of the quantization bin, $C(m) = \delta(m + 1/2)$, and then subtracting the *exact same noise* $\epsilon$ that the sender used, yielding the reconstruction $\hat{z} = C(m) - \epsilon$. This process requires synchronized pseudrandom number generators so that sender and receiver share the value of $\epsilon$. The key result is that this entire non-differentiable pipeline can be reparameterized as the simple noise-addition operation $\hat{z} = z + e$, where $e$ is an independent uniform noise term. Because the reconstruction $\hat{z}$ is simply the original signal $z$ plus independent noise, the partial derivative $\frac{\partial \hat{z}}{\partial z}$ is exactly 1, allowing unbiased gradients to flow from the receiver back to the sender (refer Fig 3).

**Learning Sparse Communication.** The DDCL framework was later extended to incentivize sparse communication. This is achieved through two additions: a variable-length code and a differentiable communication cost. The fixed code maps larger integer messages to longer bitstrings, with a special message (e.g., $m = 0$) mapping to a null message to represent not communicating. A penalty term is then added to the RL objective to encourage shorter messages. Since the true bit length is not differentiable, a surrogate cost is derived as an upper bound on the expected message length using Jensen's inequality. This results in the communication cost $\mathcal{L}_{comms} = \log_2(|M|z + 1)$ for a signal $z \in [0, 1]$, where $|M| = 1/\delta$ is the number of discrete messages. Minimizing this cost encourages the policy to output smaller-magnitude signals for $z$, which are more likely to be quantized to smaller integer values of $m$, resulting in shorter bitstrings.

While the original DDCL framework provided these innovations, it was limited to bounded, positive signals (i.e., $z \in [0, 1]$), which places unwanted architectural constraints on the policy network. In this work, we generalize this framework and its associated communication loss to handle unbounded, real-valued signals, as we detail in Section 4.

## 3 RELATED WORKS

**Multi-Agent Reinforcement Learning** (MARL) extends single-agent RL to domains where multiple agents interact in a shared environment, often modeled as a Markov game (Yang & Wang, 2021). Unlike single-agent settings, MARL introduces *non-stationarity*, since each agent's behavior affects the dynamics experienced by others (Foerster et al., 2018a). Approaches to MARL are typically categorized as either *centralized*, *decentralized*, or *centralized training with decentralized execution* (CTDE) (Amato, 2024; Oliehoek & Amato, 2016a).

In centralized MARL, a single policy chooses joint actions for all agents (Claus & Boutilier, 1998), but scaling to many agents is challenging. Decentralized methods let each agent train independently (Littman, 1994), which can be unstable in larger tasks. The CTDE framework strikes a balance by training with global information while deploying decentralized policies at execution. Value decomposition (VD) methods, such as *QMIX* (Rashid et al., 2020), *VDN* (Sunehag et al., 2017), and *QTRAN* (Son et al., 2019), factor the joint Q-function to improve credit assignment, whereas actor-critic-based methods (Lowe et al., 2017; Foerster et al., 2018b; Yu et al., 2022) learn centralized critics and decentralized actors. However, these techniques primarily rely on implicit coordination rather than explicitly modeling communication among agents.

**Communication in MARL.** To enhance coordination under partial observability, recent work explicitly incorporates *inter-agent communication*. Early *differentiable communication learning (DCL)* approaches, such as DIAL (Foerster et al., 2016a), permitted backpropagation through discrete messages using deep Q-networks. CommNet (Sukhbaatar et al., 2016a) introduced continuous channels, and IC3Net (Singh et al., 2019) provided a gating mechanism for selective message exchange, but both assume fixed topologies that can be bandwidth-inefficient. TarMAC (Das et al.,

2019) employs soft-attention for targeted message passing, while IMAC (Wang et al., 2020) leverages graph-based scheduling. On the other hand, *reinforced communication learning (RCL)* treats message selection as discrete policy optimization (Mordatch & Abbeel, 2018; Eccles et al., 2019), offering flexibility but often leading to high sample complexity.

More recent works propose dynamic communication structures via scheduling or graph optimization. For example, SchedNet (Kim et al., 2019) and MAGIC (Niu et al., 2021) use structured graphs to adaptively schedule messages, while GA-Comm (Liu et al., 2020) and CommFormer (Hu et al., 2024) learn or refine communication graphs through attention. However, these methods typically assume *continuous* messages or fixed bandwidth models, limiting real-world applicability where the channel is inherently discrete.

## 4 MULTI-AGENT RL WITH DIFFERENTIABLE DISCRETE COMMUNICATIONS

While many MARL+Comms algorithms learn to schedule messages, they typically treat the messages themselves as fixed-precision vectors. This leads to inefficient bandwidth use, as agents cannot modulate the *precision* of their messages to match the informational needs of the current context. DDCL (Freed et al., 2020c;a) provides a foundation for learning this precision, but its original formulation is limited to bounded signals ($z \in [0, 1]$), requiring restrictive activation functions (e.g., sigmoid) on policy network outputs and limiting its general applicability.

Our primary methodological contribution is to generalize the DDCL framework to support **unbounded, real-valued signals** ($z \in \mathbb{R}^d$), transforming it into a universal, "plug-and-play" layer for any MARL architecture. We achieve this because of two key properties of DDCL: the unbiased gradient estimation holds true also for unbounded signals, and the communication loss can be modified to support them. A proof of the unbiased gradient property is provided in appendix A. The loss derivation and application of our framework are presented in the following sections.

### 4.1 A DIFFERENTIABLE COMMUNICATION COST FOR UNBOUNDED SIGNALS

To encourage agents to communicate sparsely, we need a differentiable penalty on message length. We derive a new communication cost that is an upper bound on the expected bit length for an unbounded, signed integer message $m$. Our derivation adapts the approach from prior work (Freed et al., 2020c) by first assuming a variable-length code where the bit length can be upper-bounded by $length(m_b) \leq \log_2(2|m| + 1)$ to account for a sign bit. Second, we note that for our unbounded quantization scheme, the expected magnitude of the discrete message is linearly proportional to the signal's magnitude, $\mathbb{E}[|m| \mid z] = |z|/\delta$. Third, we apply Jensen's inequality to the concave logarithm function, which bounds the expected length as $\mathbb{E}[length(m_b) \mid z] \leq \log_2(2\mathbb{E}[|m| \mid z] + 1)$. Finally, by substituting the expected magnitude, we arrive at our per-dimension communication cost. Summing this over all message dimensions, agents, and timesteps gives the final objective, which sums the per-dimension cost over all agents, timesteps, and communication edges $e$:

$$\mathcal{L}_{comms} = \sum_{i=1}^{N} \sum_{t=0}^{T} \sum_{e \in E} \sum_{k=1}^{d} \log_2 \left( \frac{2|z_t^e[k]|}{\delta} + 1 \right). \tag{1}$$

In this equation, $z_t^e[k]$ denotes the $k$-th element of the signal vector. A formal proof is provided in appendix B. By minimizing this differentiable upper bound, we pressure the agent's policy to produce lower-magnitude signals for $z$, which in turn leads to shorter, sparser discrete messages.

### 4.2 FRAMEWORK APPLICATION

Our generalized DDCL serves as a simple, plug-and-play module for any MARL+Comms algorithm that uses differentiable communication channels. Incorporating it requires only inserting the DDCL quantization procedure (appendix A) into the communication channel and adding the communication cost from eq. (1) as a regularizer to the algorithm's primary loss function, weighted by a hyperparameter $\lambda$.

We apply this framework to our four baseline algorithms. We assume a star-shaped communication graph, where peripheral agents send messages to a central agent which then broadcasts an aggregated signal is inherent to IC3-Net's centralized communication design. Adhering to this structure

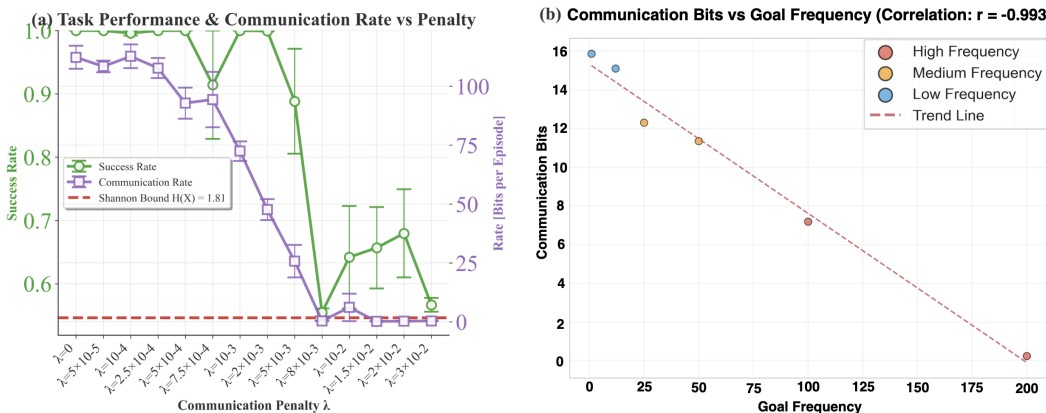

Figure 1: Qualitative analysis of the learned communication protocol in the 'Communicating-GoalEnv' toy problem. (a) Plots the Success Rate and Communication Rate against different $\lambda$ values. The episodic plot illustrates a "lossless compression" regime where the success rate remains perfect (1.0) while the required communication bits are significantly reduced as $\lambda$ increases to $8 \times 10^{-3}$. (b) A per-timestep comparison of the learned communication policy with the ground-truth goal sampling frequency. The strong negative correlation (r=-0.993) demonstrates that the agent learns a frequency-aware code, allocating the fewest bits to the most probable goals.

allows us to isolate the specific impact of our adaptive precision mechanism on the architecture. For TarMAC and GA-Comm, which operate on a fully-connected graph, DDCL is applied to every communication round within their respective attention blocks. This involves two message-passing steps per block: first, when agents broadcast their key vectors to all other agents, and second, when the resulting aggregated value vectors are communicated back to each agent. The hard-attention mechanisms in these models serve to prune the graph, and DDCL then operates on the remaining connections. Similarly, for MAGIC, which uses multiple graph attention-based scheduling networks, DDCL is applied to the message-passing within each of its attention mechanisms.

## 5 EXPERIMENTS

Our experiments are designed to validate our three primary contributions. First, in a controlled setting, we provide a qualitative analysis to build intuition for *how* our generalized DDCL learns to modulate communication precision. Second, we demonstrate DDCL's practical utility and generality by integrating it into a wide range of state-of-the-art MARL+Comms algorithms across several challenging benchmarks. Finally, we provide direct evidence for the "Bitter Lesson" in MARL communication by showing that a simple, general-purpose architecture empowered by DDCL can compete with complex, bespoke communication models.

### 5.1 QUALITATIVE ANALYSIS OF AN EMERGENT COMMUNICATION PROTOCOL

To build intuition for *how* our generalized DDCL framework enables agents to learn efficient communication, we first conduct a qualitative analysis in a controlled, interpretable environment. Our goal is to move beyond simply measuring bandwidth reduction and instead investigate the structure of the learned communication protocol itself.

**Experimental Setup.** We use a simple 8x8 grid-world task, `CommunicatingGoalEnv`, with a stationary "speaker" and a mobile "listener". The information is asymmetric to create a clear communication bottleneck: the speaker observes the raw integer (x,y) coordinates of the goal, while the listener observes its own integer (x,y) coordinates. At the start of each episode, a goal is sampled from a non-uniform distribution defined by the probabilities: '(0,0):51.5%, (7,7):25.8%, (3,4):12.9%, (4,3):6.4%, (1,6):3.1%, (6,1):0.3%'. The agents are trained jointly using Multi-Agent Proximal Policy Optimization (MAPPO) (Yu et al., 2022) to maximize a cooperative reward for reaching the goal, while the speaker's policy is simultaneously penalized by our communication

cost $\mathcal{L}_{comms}$ (Eq. 1) with a coefficient of $\lambda_{comms} = 4e-3$. The trained policy achieves a 100% success rate, allowing our analysis to focus purely on the efficiency of the learned communication protocol. An optimal protocol, akin to a compression algorithm, should assign the shortest bitstrings to the most frequent goals.

**Agents Learn a Frequency-Aware Communication Protocol.** Our analysis reveals that the agent learns a sophisticated, frequency-aware communication protocol, a key principle of efficient coding. As shown in fig. 1(b), there is a strong negative correlation of **r = -0.989** between a goal's frequency and the number of bits the agent allocates to communicate it. This learned, variable-length encoding is highly specialized: the agent allocates just **0.25 bits** on average to communicate the most frequent goal (occurring 51.5% of the time), while systematically assigning higher costs to rarer events, culminating in **15.98 bits** for the least frequent one (0.3% frequency). **This vast difference in allocated bits directly reflects the agent's training experience, where it observes the most common goal in over 25,000 episodes but encounters the rarest goal in only 52 episodes.** This quantitative analysis reveals the protocol's sophistication: the agent correctly identifies that the most frequent goal is over 170 times more likely to occur than the rarest one, and in response, it develops a code that is nearly **64 times more bit-efficient** for this common event. This learned strategy provides a dramatic advantage where it matters most. Compared to the de facto standard in MARL—a fixed-precision uniform code that would require $\lceil \log_2(64) \rceil = 6$ bits for any goal—our agent is **24 times more efficient** when communicating the most likely event.

**Analysis of Divergence from the Theoretical Optimum.** While the learned protocol is highly effective, its average message length of **4.75 bits** is greater than the theoretical minimum defined by the Shannon Entropy of the source distribution. The entropy, $H(X) = -\sum_i p_i \log_2(p_i)$, is calculated from the goal probabilities as $H(X) = -(0.515 \times \log_2(0.515) + \cdots + 0.3 \times \log_2(0.3)) \approx 1.81$ bits. This divergence is an expected and informative consequence of our optimization framework, stemming from three primary factors.

First, the agent minimizes a differentiable surrogate, $\mathcal{L}_{comms}$, which is an upper bound on the true expected message length. This bound is derived using Jensen's inequality

$$\mathbb{E}[\log_2(2|m| + 1)] \leq \log_2(2\mathbb{E}[|m|] + 1).$$

The agent minimizes the term on the right-hand side. Minimizing an upper bound does not guarantee the minimization of the original value; the "Jensen gap" between the two sides introduces slack into the optimization problem.

Second, our loss, $\mathcal{L}_{comms}(z) = \log_2(\frac{2|z|}{\delta} + 1)$, directly couples communication cost to the L1-norm of the latent vector $z$. An information-theoretically optimal code decouples a symbol's identity from its codeword length, caring only about its probability. Our framework must learn a more complex, indirect mapping: the policy network must learn to map high-probability inputs (frequent goals) to low-magnitude latent vectors $z$. While our results show the agent is successful at this, the indirect mechanism is inherently less direct than optimizing an entropy-based objective.

Finally, the reinforcement learning process itself influences the outcome. For high-frequency goals, the speaker receives abundant learning signals to compress its message representation effectively; for instance, the agent has over **25,000 episodes** to optimize the encoding for the most frequent goal, but only **52 episodes** to learn a representation for the rarest one. This data scarcity for rare events means the speaker does not receive enough data to learn an optimally compressed representation and therefore defaults to a higher-cost, but safe and unambiguous, encoding to ensure task success. This explains the high bit cost for low-frequency goals, which contributes to the overall divergence from the Shannon limit. The final learned protocol represents a trade-off: a code that is not only efficient where it matters most, but is also simple enough for a finite-capacity network to learn and use reliably to achieve perfect task success under the data distribution it experienced.

**The Rate-Distortion Frontier.** The hyperparameter $\lambda$ serves as a predictable lever to navigate the trade-off between task performance and communication efficiency, effectively tracing out the Rate-Distortion frontier shown in fig. 1(a) of the Toy problem. A quantitative analysis of the curve reveals distinct operational regimes. For small values of $\lambda$ (e.g., $10^{-5}$), the penalty is negligible, and the agent achieves near-perfect performance (Distortion $\approx 0$) at a high communication cost of approximately 110 bits per episode. As $\lambda$ increases to $5 \times 10^{-4}$, the agent enters a "lossless compression" phase, cutting its communication rate to ~90 bits while maintaining a perfect success rate. Increasing the penalty further forces the agent into "lossy" compression; at $\lambda = 8 \times 10^{-3}$, the

rate is reduced to just a few bits, but at a steep cost, with distortion rising sharply to approximately 0.45 (a success rate of only 55%). Qualitatively, this demonstrates that the framework first learns to discard redundant information. Only when the communication penalty becomes critically high does it begin to discard information essential for coordination, causing a sharp decline in task success.

## 5.2 General Utility on MARL Benchmarks

Next, we demonstrate that our generalized DDCL is a robust, plug-and-play module that improves communication efficiency across a wide range of complex MARL algorithms and challenging environments.

**MARL Environments.** We evaluate our approach on three diverse and standard benchmarks. First, we use **Traffic Junction (TJ)** (Singh et al., 2019), a cooperative navigation task where agents must avoid collisions in Medium (10 agents) and Hard (20 agents) settings. Second, we use **Predator-Prey (PP)** (Singh et al., 2019), a cooperative pursuit task where predators coordinate to capture prey in Medium (5 predators) and Hard (10 predators) settings. Finally, we test on the high-dimensional, physics-based **Google Research Football (GRF)** environment (Singh et al., 2019), evaluating on a 3-vs-1 attacking scenario and a corner kick setup with sparse rewards. For all tasks, performance is measured by the episode success rate. Details about the environment in appendix E

**Baseline Algorithms and Evaluation Protocol.** We integrate DDCL into four state-of-the-art MARL+Comms algorithms that feature different, specialized communication architectures: **IC3Net** (Singh et al., 2019), which uses a learned binary gate; **TarMAC** (Das et al., 2019), which employs soft attention; **GAComm** (Liu et al., 2020), which uses graph attention; and **MAGIC** (Niu et al., 2021), which separates message scheduling and processing. For each baseline, we compare the original implementation (using 32-bit float messages), several versions using fixed quantization (detailed in appendix F) with a Straight-Through Estimator (STE), and our DDCL version.

**DDCL as a Universal Efficiency Multiplier.** Our results consistently show that DDCL serves as a universal efficiency multiplier, dramatically improving the performance-bandwidth trade-off, consistently reducing communication by one to five orders of magnitude across all tested architectures and environments. The Pareto plots in fig. 2 illustrate this clearly: most DDCL-enhanced variants (red markers) consistently form the global Pareto frontier (indicated by thick black borders), representing the best achievable performance for a given communication budget. Crucially, this efficiency gain frequently enables superior performance. The most transformative result is seen with IC3Net in the complex GRF 3v1 environment, where DDCL achieves a statistically significant 467.1% increase in success rate, learning a more effective coordination protocol than its high-precision counterpart. While the high stochasticity of the most complex tasks leads to wide confidence intervals for many runs, the mean performance shows substantial improvements across the board, such as for TarMAC (+49.9% gain) and MAGIC (+176.3% gain) in different settings. The few instances where a performance trade-off occurs highlight DDCL's ability to navigate the efficiency-performance frontier, sacrificing a small amount of performance for immense ($>$7000x) communication savings (MAPPO).

DDCL's adaptive precision proves to be a fundamentally more robust and powerful strategy than naive, fixed-quantization (STE). This is most evident in challenging environments like Predator-Prey Hard, where DDCL enables the MAGIC and GAComm agents to achieve significant success rate gains of over 155% and 75% respectively, compared to their STE counterparts. This highlights that in complex coordination problems, simply using fewer bits is not enough; learning how to communicate precisely is critical. While DDCL is the superior approach in the vast majority of cases, we identify a few specific task-architecture pairings where a simple STE baseline proves competitive (e.g., TarMAC in TJ_Hard). These rare exceptions are valuable findings that reveal complex local optima in the learning landscape. However, the overwhelming trend confirms that DDCL's learned, adaptive approach provides a more reliable path to high-performing, communication-efficient agents. For a detailed breakdown of these results, please see Appendix G.

## 5.3 Validating the "Bitter Lesson"

Our final and broadest contribution is to test the hypothesis that the "Bitter Lesson" applies to MARL communication: general-purpose methods that leverage computation are ultimately superior to those relying on complex, human-designed priors.

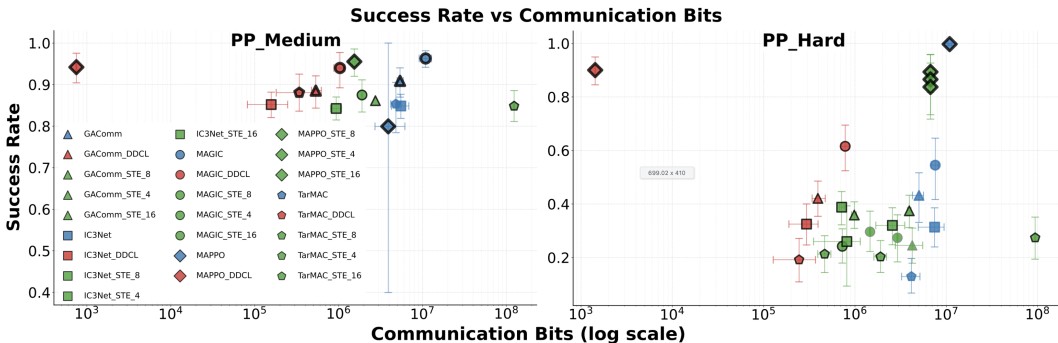

(a) In **Predator-Prey**, DDCL variants (red) consistently establish the Pareto frontier, often improving the success rate (e.g., TarMAC in Hard) while using orders of magnitude less bandwidth than original (blue) and STE (green) baselines.

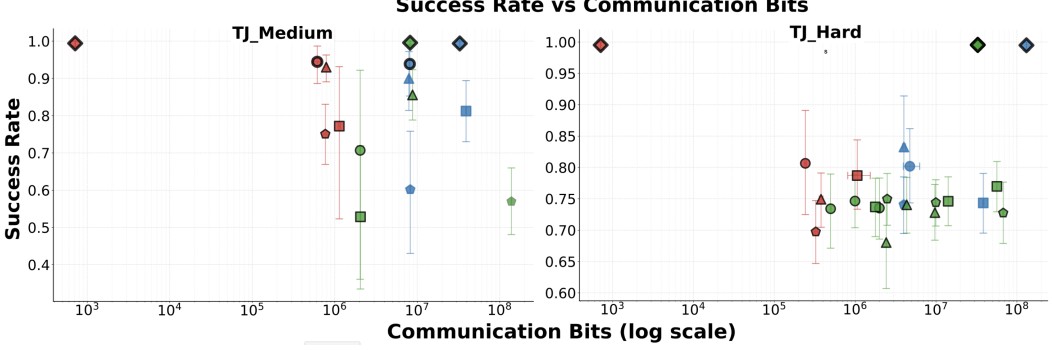

(b) In **Traffic Junction**, results are nuanced. Variants like MAPPO achieve extreme lossless compression, while others trade a small amount of performance for significant efficiency gains, consistently outperforming naive STE quantization.

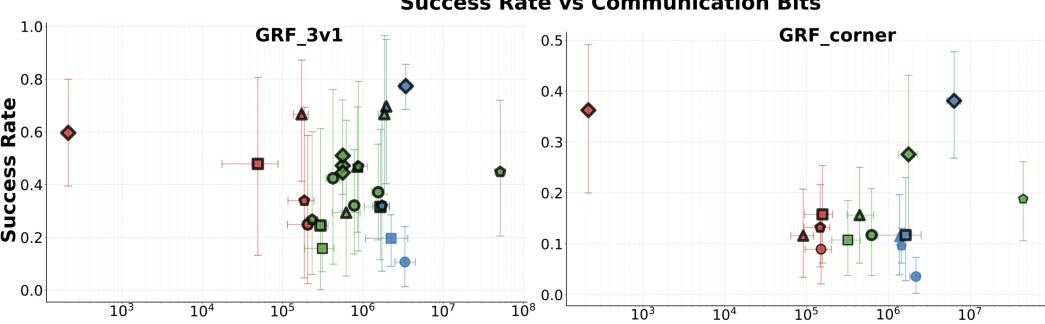

(c) In the complex **Google Research Football** environment, DDCL enables transformative gains, with architectures like IC3Net more than doubling their success rate, highlighting its ability to learn coordination in high-dimensional, sparse-reward tasks.

Figure 2: **Performance versus communication bandwidth across all benchmark environments.** Each point represents an algorithm variant's mean performance (Success Rate) and communication cost (Bits per episode, log scale) over 5 different seeds. Error bars denote 95% confidence intervals. Note that Y-axis scales are often focused on a specific range and may not start at zero. The top-left of each plot represents the ideal outcome (high success, low communication cost), while the bottom-right is the worst. Our DDCL-enhanced variants (red markers) consistently operate on the left side of the plots, demonstrating significant communication savings. The global Pareto frontier, representing the best possible trade-offs, is marked with a **thick black border**, while algorithm-specific frontiers are marked with a **thin black border**. 'STE_X' refers to a baseline using a Straight-Through Estimator to quantize 32-bit float messages to 'X' bits.

**Experimental Design.** We test this hypothesis by comparing a simple, general architecture against the complex, specialized baselines from the previous section. Our model, dubbed **MAPPO_Transformer** (we drop Transformer for all variants from the naming in our figures for brevity), uses MAPPO algorithm with encoder-only Transformer networks for the policy. This architecture serves as a powerful, general-purpose graphical data processor (Joshi, 2025) without any bespoke communication mechanisms. Communication is achieved simply by applying our DDCL framework to message vectors (keys and aggregated attention value vectors). We then train this simple "MAPPO_Transformer_DDCL" agent and compare its position on the performance-vs-bandwidth Pareto frontier against all the specialized baselines across all environments.

**Generality and Scale Outperform Bespoke Design.** Our findings provide strong evidence for the "Bitter Lesson" in MARL communication. As illustrated across all benchmarks in fig. 2, the simple MAPPO agent, when empowered by DDCL, achieves a performance-bandwidth trade-off that is highly competitive with, and often superior to, the more complex, specialized architectures. In nearly every environment, the MAPPO_Transformer_DDCL variant lies on the global Pareto frontier, demonstrating that it is among the best-performing models regardless of communication budget. For example, in Predator-Prey Hard, it achieves a higher success rate than all other models while using orders of magnitude less communication.

This result has significant implications. It suggests that the intricate, hand-crafted gating, scheduling, and graph-attention mechanisms of the specialized baselines may be less critical than the foundational ability to learn communication precision. By combining a general, scalable architecture (the Transformer) with a general, learnable communication regularizer (DDCL), we can match or exceed the performance of bespoke systems. This indicates that the core competency of efficient communication can be learned through general-purpose methods that leverage large-scale training, rather than being explicitly engineered through human-designed architectural priors.

We conducted an extensive sensitivity analysis on the two primary hyperparameters of our DDCL framework: the communication cost coefficient, $\lambda$, and the quantization granularity, $\delta$ in Appendix I.

## 6 LIMITATIONS AND FUTURE WORK

While our work demonstrates that DDCL is a powerful and general framework, we acknowledge several limitations that also chart exciting paths for future research. The current quantization grid is uniform and fixed; future work could learn a **per-channel granularity** ($\delta_k$) or explore **non-uniform grids** via learnable noise distributions (e.g., Gaussians). Furthermore, the current communication loss couples signal magnitude to bitrate; a more principled **entropy-based loss** could decouple this by learning a code based on signal frequency rather than magnitude. Finally, the practical requirement of **shared randomness** between agents is a key constraint; investigating the framework's robustness to desynchronized noise would improve its real-world applicability.

## 7 CONCLUSION

In this work, we presented a generalized DDCL framework, transforming it into a universal, plug-and-play module for learning efficient communication in any MARL architecture. By extending the core mechanism to support unbounded, signed signals, we removed architectural constraints that previously limited its applicability. Our experiments provide a threefold validation of our approach. Our qualitative analysis revealed that agents learn a near-optimal, frequency-aware communication protocol by assigning the shortest, lowest-cost messages to the most frequent events. We then demonstrated the framework's general utility by integrating it into several state-of-the-art MARL algorithms, showing it can reduce communication bandwidth by orders of magnitude while maintaining or improving task performance. Finally, we provided strong evidence for the "Bitter Lesson" in MARL communication by showing that a simple, general-purpose Transformer agent empowered by DDCL can match and often exceed the performance of complex, specialized architectures. Taken together, our findings challenge the prevailing trend of designing intricate, hand-crafted communication modules, suggesting that a more promising research direction lies in combining general, scalable architectures with principled, learnable mechanisms for efficient communication. The limitations of our current approach, which we detailed in the previous section, chart a clear path for future work in this exciting direction.

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

## A  PROOF OF STATISTICAL INDEPENDENCE BETWEEN THE QUANTIZATION ERROR $e$ AND THE SIGNAL $z$

In this section, we prove the critical property that the reconstruction error $e$ is statistically independent of the original signal $z$. We establish this result through two distinct but complementary approaches. The first proof provides a geometric interpretation theorem A.1 to build intuition, while the second offers a more formal, analytical derivation to rigorously confirm the finding theorem A.2.

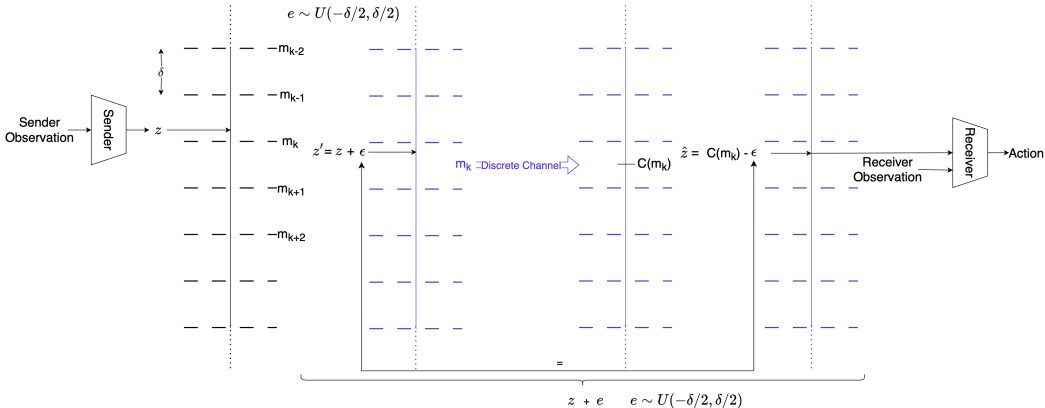

Figure 3: An overview of the generalized DDCL procedure. A sender's unbounded, real-valued signal $z$ is perturbed, quantized, and sent as a discrete message $m$. The receiver uses shared randomness to reconstruct the signal $\hat{z}$ in a way that allows gradients to flow back to the sender.

**Theorem A.1** (Statistical Independence of Quantization Error $e$ and communication vector $z$). *Let $z \in \mathbb{R}$ be a real-valued signal and let the noise $\epsilon$ be a random variable drawn from a uniform distribution, $\epsilon \sim U(-\delta/2, +\delta/2)$. Let the signal be quantized by first producing a noisy signal $z' = z + \epsilon$, which is then used to generate a reconstructed signal $\hat{z} = (\lfloor z'/\delta \rfloor + 1/2)\delta - \epsilon$. Let the reconstruction error be defined as $e = \hat{z} - z$.*

*Then, the reconstruction error $e$ is statistically independent of the signal $z$ and is uniformly distributed, $e \sim U(-\delta/2, +\delta/2)$.*

*Proof of Error Independence via Geometric Interpretation.* The objective is to prove that for any real-valued signal $z \in \mathbb{R}$, the reconstruction error $e = \hat{z} - z$ is statistically independent of $z$ and follows a uniform distribution, $e \sim U(-\delta/2, +\delta/2)$.

First, we define the components of the communication procedure. The sender begins with a signal $z \in \mathbb{R}$, which is perturbed by noise $\epsilon \sim U(-\delta/2, +\delta/2)$ to produce a noisy signal $z' = z + \epsilon$. This signal is then quantized to an integer message $m = \lfloor \frac{z'}{\delta} \rfloor$. The center of the quantization bin containing $z'$ is given by the function

$$C(m) = \left( \left\lfloor \frac{z'}{\delta} \right\rfloor + \frac{1}{2} \right) \delta. \tag{2}$$

$m$ is the integer bin number, representing the discrete message sent over the channel. It is calculated by rounding the noisy signal $z'$ to the nearest quantization grid point.

$C(m)$ is the quantization point, which is the real-valued center of the bin identified by $m$. It serves as the representative value for that bin and is the starting point for the receiver's signal reconstruction process.

Finally, the receiver reconstructs the signal as $\hat{z} = C(m) - \epsilon$.

From these definitions, we derive the expression for the reconstruction error $e = \hat{z} - z$. By substituting $z = z' - \epsilon$ and the definition of $\hat{z}$, we find

$$e = (C(m) - \epsilon) - (z' - \epsilon)$$
$$e = C(m) - z'.$$

This result reveals that the reconstruction error $e$ is exactly the quantization error of the noisy signal $z'$ (assuming $C(m)$ as the quantization point), namely the difference between $z'$ and the center of its quantization bin.

The event that the error $e$ takes a specific value $y$ is equivalent to the condition $C(m) - z' = y$. The bin center $C(m)$ must be a point on the quantization grid, so $C(m) = (k + \frac{1}{2})\delta$ for some integer $k \in \mathbb{Z}$. Substituting this, the error event occurs if, for some integer $k$,

$$(k + \frac{1}{2})\delta - z' = y, \tag{3}$$

which is equivalent to $z'$ taking a value from the set $\{(k + \frac{1}{2})\delta - y \mid k \in \mathbb{Z}\}$.

The conditional probability density function (PDF) of the error $e$ is the sum of the probabilities of $z'$ taking on any of these specific values:

$$p(e = y \mid z) = \sum_{k \in \mathbb{Z}} p\left(z' = \left(k + \frac{1}{2}\right)\delta - y \mid z\right). \tag{4}$$

The PDF of $z'$ is a uniform distribution over the interval $[z - \delta/2, z + \delta/2]$. A term in the sum from eq. (4) is non-zero only if the point $(k + \frac{1}{2})\delta - y$ lies within this interval:

$$z - \frac{\delta}{2} \leq \left(k + \frac{1}{2}\right)\delta - y \leq z + \frac{\delta}{2}. \tag{5}$$

We now prove that for any $z \in \mathbb{R}$ and any valid error $y \in [-\delta/2, +\delta/2]$, there is exactly one integer $k$ that satisfies this condition. Solving for $k$ yields

$$\frac{z}{\delta} - 1 + \frac{y}{\delta} \leq k \leq \frac{z}{\delta} + \frac{y}{\delta}.$$

This defines a closed interval for the integer $k$. The length of this interval is

$$\text{Length} = \left(\frac{z}{\delta} + \frac{y}{\delta}\right) - \left(\frac{z}{\delta} - 1 + \frac{y}{\delta}\right) = 1. \tag{6}$$

A closed interval of real numbers with length 1 contains exactly one integer. Therefore, a unique integer solution, which we denote $k^*$, exists for any $z$ and $y$.

Because a unique solution $k^*$ exists, the infinite sum in eq. (4) collapses to a single non-zero term:

$$p(e = y \mid z) = p\left(z' = \left(k^* + \frac{1}{2}\right)\delta - y \mid z\right). \tag{7}$$

Since this point lies within the support of the uniform distribution of $z'$, the value of the PDF is constant and equal to $1/\delta$. Therefore,

$$p(e = y \mid z) = \frac{1}{\delta}. \tag{8}$$

The resulting conditional density does not depend on the original signal $z$. Thus, the reconstruction error $e$ is statistically independent of the signal $z$, and its distribution is that of a uniform random variable, $U(-\delta/2, +\delta/2)$. □

**Theorem A.2** (Statistical Independence Between $e$ and $z$.). *Assume the stochastic quantization strategy described in Sec. 4: let $z \in \mathbb{R}$, and $\epsilon \sim U(\frac{-\delta}{2}, \frac{+\delta}{2})$. Messages are quantized into discrete message $m$ based on which uniformly spaced quantization intervals $z' = z + \epsilon$ falls into, i.e., message $m$ is chosen iff $L(m) \leq z + \epsilon < U(m)$, where $U(m)$ and $L(m)$ are the upper and lower bounds of the quantization interval corresponding to $m$, respectively, and $U(m) - L(m) = \delta$. The "reconstructed" $z$ is given by $\hat{z} = C(m) - \epsilon$, where $C(m)$ is the center of the quantization interval corresponding to $m$. Then we have that $e \sim U(-\frac{\delta}{2}, \frac{\delta}{2})$, and $e \perp\!\!\!\perp z$.*

*Proof.* We start by expressing $P(e|z)$ in terms of the full joint distribution over $e$, $m$, and $\epsilon$ given $z$, marginalized over $m$ and $\epsilon$.

$$P(e|z) = \sum_m \int_{-\infty}^{+\infty} P(e|z, m, \epsilon) P(m|z, \epsilon) P(\epsilon) d\epsilon, \tag{9}$$

$$= \frac{1}{\delta} \sum_m \int_{-\frac{\delta}{2}}^{+\frac{\delta}{2}} P(e|z, m, \epsilon) P(m|z, \epsilon) d\epsilon. \tag{10}$$

We note that for a given $z$ there are only two possible values of $m$, which we denote $m_a$ and $m_b$, where $m_a$ is the lower of the two possible intervals, such that $L(m_a) \leq z - \frac{\delta}{2} < U(m_b)$, and $m_b$ is the higher of the two, such that $L(m_b) \leq z + \frac{\delta}{2} < U(m_b)$. Eliminating all other terms from the above summation, we have

$$P(e|z) = \frac{1}{\delta} \left( \int_{-\frac{\delta}{2}}^{+\frac{\delta}{2}} P(e|z, m_a, \epsilon) P(m_a|z, \epsilon) d\epsilon + \int_{-\frac{\delta}{2}}^{+\frac{\delta}{2}} P(e|z, m_b, \epsilon) P(m_b|z, \epsilon) d\epsilon \right). \tag{11}$$

By our definition of $m_a$ and $m_b$, we have that

$$P(m_a|z, \epsilon) = \begin{cases} 1 & \text{if} \quad L(m_a) - z \leq \epsilon < U(m_a) \\ 0 & \text{otherwise,} \end{cases} \tag{12}$$

and

$$P(m_b|z, \epsilon) = \begin{cases} 1 & \text{if} \quad L(m_b) - z \leq \epsilon < U(m_b) \\ 0 & \text{otherwise.} \end{cases} \tag{13}$$

We can therefore further restrict the bounds of the integral over $\epsilon$ to include only regions where $P(m_a|z, \epsilon)$ and $P(m_b|z, \epsilon)$ are nonzero, respectively:

$$P(e|z) = \frac{1}{\delta} \left( \int_{-\frac{\delta}{2}}^{U(m_a)-z} P(e|z, m_a, \epsilon) P(m_a|z, \epsilon) d\epsilon + \int_{L(m_b)-z}^{+\frac{\delta}{2}} P(e|z, m_b, \epsilon) P(m_b|z, \epsilon) d\epsilon \right). \tag{14}$$

Because the error is deterministic given $m$, $\epsilon$, and $z$, and is given by $e = C(m) - \epsilon - z$, we have that $P(e|z, m_a, \epsilon) = \delta(C(m) - \epsilon - z)$, where $\delta$ is a Dirac delta function. Evaluating the above integral, we have

$$P(e|z) = \frac{1}{\delta} \left( \begin{cases} 1 & \text{if} \quad \epsilon \in [\frac{-\delta}{2}, U(m_a) - z) \\ 0 & \text{otherwise} \end{cases} + \begin{cases} 1 & \text{if} \quad \epsilon \in [L(m_b), \frac{\delta}{2}) \\ 0 & \text{otherwise} \end{cases} \right) \tag{15}$$

$$= \frac{1}{\delta} \left( \begin{cases} 1 & \text{if} \quad \epsilon \in [\frac{-\delta}{2}, U(m_a) - z) \\ 0 & \text{otherwise} \end{cases} + \begin{cases} 1 & \text{if} \quad \epsilon \in [U(m_a), \frac{\delta}{2}) \\ 0 & \text{otherwise} \end{cases} \right) \tag{16}$$

$$(\text{Because } U(m_a) = L(m_b)) \tag{17}$$

$$= \frac{1}{\delta} \left( \begin{cases} 1 & \text{if} \quad \epsilon \in [\frac{-\delta}{2}, \frac{\delta}{2}] \\ 0 & \text{otherwise} \end{cases} \right) \tag{18}$$

$$= U(-\frac{\delta}{2}, \frac{\delta}{2}). \tag{19}$$

Thus we have that $e$ is uniformly distributed and independent of $z$.

$\square$

## B  DERIVATION OF THE DIFFERENTIABLE COMMUNICATION COST

In this section, we provide a detailed derivation of the differentiable communication cost, $\mathcal{L}_{comms}$. The objective is to find a differentiable function that serves as a tight upper bound for the expected bit length of a message, $\mathbb{E}[\text{length}(m_b) \mid z]$, given the continuous, unbounded signal $z \in \mathbb{R}$. The derivation proceeds in three main steps.

*Proof.* We first establish an information-theoretic upper bound on the number of bits required to transmit a signed integer message $m \in \mathbb{Z}$. A common and efficient method to encode a signed integer is to first map it to a unique non-negative integer. A standard mapping is $f(m) = 2|m|$ if $m \geq 0$ and $f(m) = 2|m| - 1$ if $m < 0$. The largest value this mapping can produce for a given magnitude $|m|$ is $2|m|$. The number of bits required to encode a non-negative integer $k$ is approximately $\log_2(k + 1)$. Therefore, the bit length of our encoded message $m_b$ can be upper-bounded by the cost of encoding the largest possible value, $2|m|$:

$$\text{length}(m_b) \leq \log_2(2|m| + 1). \tag{20}$$

This bound gracefully handles $m = 0$ and grows logarithmically with the magnitude of the message, effectively encoding both sign and magnitude information in a single expression.

Next, we establish the relationship between the expected magnitude of the discrete message, $\mathbb{E}[|m| \mid z]$, and the magnitude of the original continuous signal, $|z|$. The message $m$ is generated by rounding the noisy signal $z' = z + \epsilon$:

$$m = \left\lfloor \frac{z + \epsilon}{\delta} + \frac{1}{2} \right\rfloor. \tag{21}$$

Since the noise $\epsilon$ is drawn from a zero-mean distribution $U(-\delta/2, +\delta/2)$, the rounding operation is symmetric around $z/\delta$. Therefore, the expected value of the message is $\mathbb{E}[m \mid z] = z/\delta$. For the purpose of deriving a tractable surrogate loss, we assume that the expectation of the magnitude is approximately the magnitude of the expectation. This assumption is well-justified when $|z| \gg \delta$.

$$\mathbb{E}[|m| \mid z] = \frac{|z|}{\delta}. \tag{22}$$

This linear relationship forms the basis of our differentiable surrogate.

Our goal is to find a differentiable upper bound on $\mathbb{E}[\text{length}(m_b) \mid z]$. We begin by taking the expectation of the inequality from equation 20:

$$\mathbb{E}[\text{length}(m_b) \mid z] \leq \mathbb{E}[\log_2(2|m| + 1) \mid z]. \tag{23}$$

The function $f(x) = \log_2(x)$ is a concave function for $x > 0$. For any concave function, Jensen's inequality states that $\mathbb{E}[f(X)] \leq f(\mathbb{E}[X])$. Applying this inequality to our expression, we get:

$$\mathbb{E}[\log_2(2|m| + 1) \mid z] \leq \log_2\left(\mathbb{E}[2|m| + 1 \mid z]\right)$$
$$= \log_2\left(2\mathbb{E}[|m| \mid z] + 1\right),$$

where the second line follows from the linearity of expectation.

By chaining these results, we have an upper bound on the expected length in terms of the expected message magnitude.

$$\mathbb{E}[\text{length}(m_b) \mid z] \leq \log_2\left(2\mathbb{E}[|m| \mid z] + 1\right). \tag{24}$$

Finally, we substitute our linear magnitude assumption from equation 22 to arrive at our differentiable upper bound for a single dimension of the signal:

$$\mathbb{E}[\text{length}(m_b) \mid z] \leq \log_2\left(\frac{2|z|}{\delta} + 1\right). \tag{25}$$

This final expression is differentiable with respect to $|z|$ and serves as our surrogate communication cost. Summing this cost over all $d$ dimensions of the message vector, all communication edges $e \in E$, all $N$ agents, and all $T$ timesteps gives the final loss function:

$$\mathcal{L}_{comms} = \sum_{i=1}^{N} \sum_{t=0}^{T} \sum_{e \in E} \sum_{k=1}^{d} \log_2\left(\frac{2|z_t^e[k]|}{\delta} + 1\right). \tag{26}$$

$\square$

## C  ADDITIONAL DETAILS FOR QUALITATIVE ANALYSIS

**To supplement the analysis in Section 5.1**, we provide additional visualizations of the learned communication protocol from the `CommunicatingGoalEnv` experiment. These figures offer a more detailed view of how the agent allocates communication bits across the entire state space and for the specific, non-uniform goal distribution.

Figure 4 visualizes the learned communication policy by comparing the heatmap of bit costs against the underlying goal sampling distribution. The learned policy from our method (left) is spatially coherent; it correctly allocates the fewest bits to the most frequent goal at '(0,0)' and generalizes by assigning progressively more bits to locations as their distance from this high-frequency zone increases. The goal sampling frequency (right) shows the non-uniform probability distribution the agent was trained on. The strong visual correlation between the two heatmaps demonstrates that the agent has successfully learned the task's probability structure. This occurs because our framework's loss function couples the bit cost to the latent vector's magnitude, which encourages the network to learn a smooth and generalizable mapping from the coordinate space of the goals to the cost of communicating them.

Figure 5 provides an alternative view of the data presented in fig. 1 in the main text. It shows the communication bits for the six specific goals sampled during training, sorted by their frequency category. This visualization makes it easy to compare the discrete bit levels assigned to each goal, clearly showing the inverse relationship between goal frequency and message length, from 0.25 bits for the 'High' frequency goal to over 15.95 bits for the 'Low' frequency goals.

## D  DETAILED ANALYSIS OF THE COMMUNICATION PENALTY VIA SHANNON GAP

This section provides a detailed breakdown of the hyperparameter sweep over the communication penalty, $\lambda$, which generated the points for the Rate-Distortion curve in Section 5.1 of the main text. The results, presented in fig. 6, illustrate how $\lambda$ directly controls the communication efficiency.

**The Shannon Gap.** Figure 6 quantifies the communication inefficiency by plotting the "Shannon Gap"—the difference between the empirical rate $R(\lambda)$ and the theoretical Shannon bound of 1.81 bits. For low values of $\lambda$, the agent uses a large number of excess bits (a gap of approx 110 bits) to guarantee high performance and robustness. As the penalty increases, the agent is forced to become more efficient, and the Shannon Gap narrows dramatically. For $\lambda \geq 8 \times 10^{-3}$, the agent's communication rate drops below the theoretical minimum required to solve the task perfectly, which is reflected in the corresponding performance drop fig. 1(a). This highlights the fundamental tension between pure communication efficiency and the robustness required for a downstream RL task.

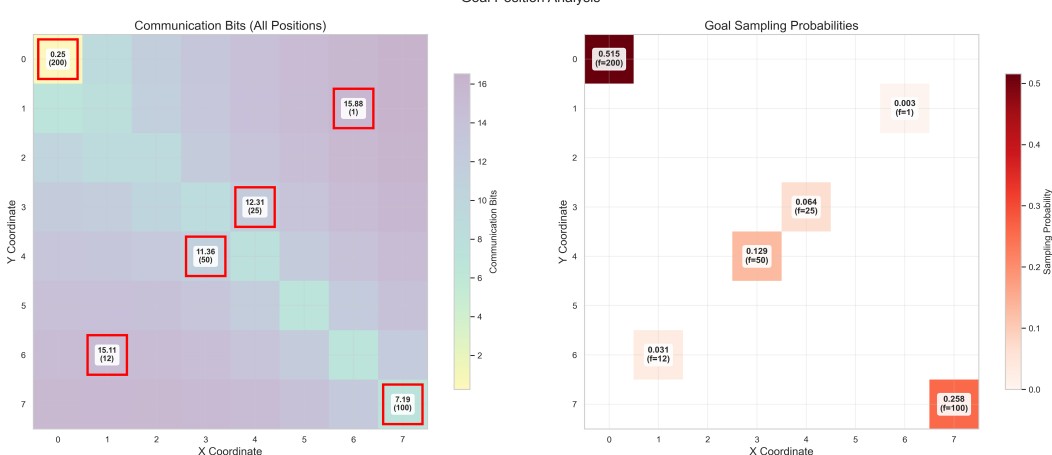

Figure 4: A comparison of the learned communication bit costs (left) with the goal sampling frequency (right) in the 8x8 grid. Our method learns a spatially smooth code that mirrors the underlying probability distribution, assigning the lowest cost to the most frequent goal at '(0,0)' and progressively higher costs to locations further away. Grid coordinates that were not sampled as goals are shown in white.

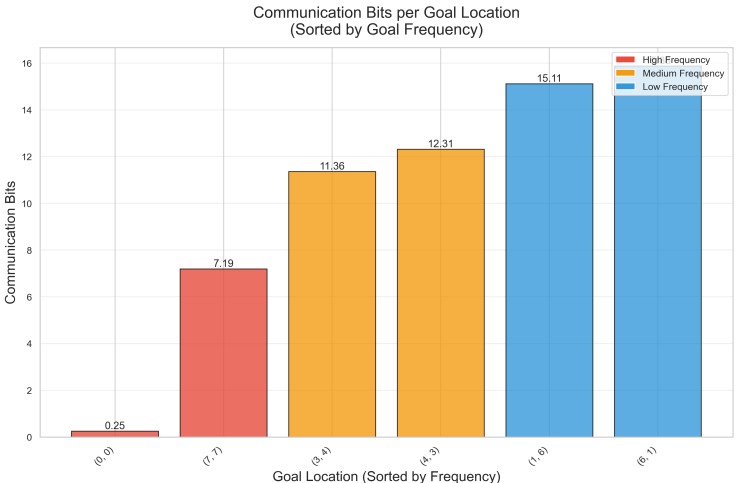

Figure 5: Communication bits per goal, sorted by the goal's frequency category. This chart clearly illustrates the learned inverse relationship: high-frequency goals are encoded with very few bits, while low-frequency goals require significantly more.

## E    DETAIL DESCRIPTION ABOUT THE ENVIRONMENT.

The **Predator-Prey (PP)** environment is a partially observable, cooperative multi-agent pursuit task set in a 'dim' x 'dim' grid world. At the start of each episode, a configurable number of *predator* agents and many stationary *prey* are placed at random, non-overlapping locations on the grid. The objective is for all predators to coordinate and simultaneously occupy the same grid cell as the prey to achieve a successful capture. The task is partially observable, as each predator's observation is limited to a local 'vision' grid centered on itself. This observation is a one-hot encoded tensor that identifies the locations of other predators and the prey within its field of view. Each predator has a discrete action space of five actions: *UP*, *DOWN*, *LEFT*, *RIGHT*, and *STAY*. The reward structure is designed to encourage coordinated and efficient captures: agents receive a small penalty at each timestep ('-0.05'), and a positive reward is given when a predator is on the prey's cell, with the

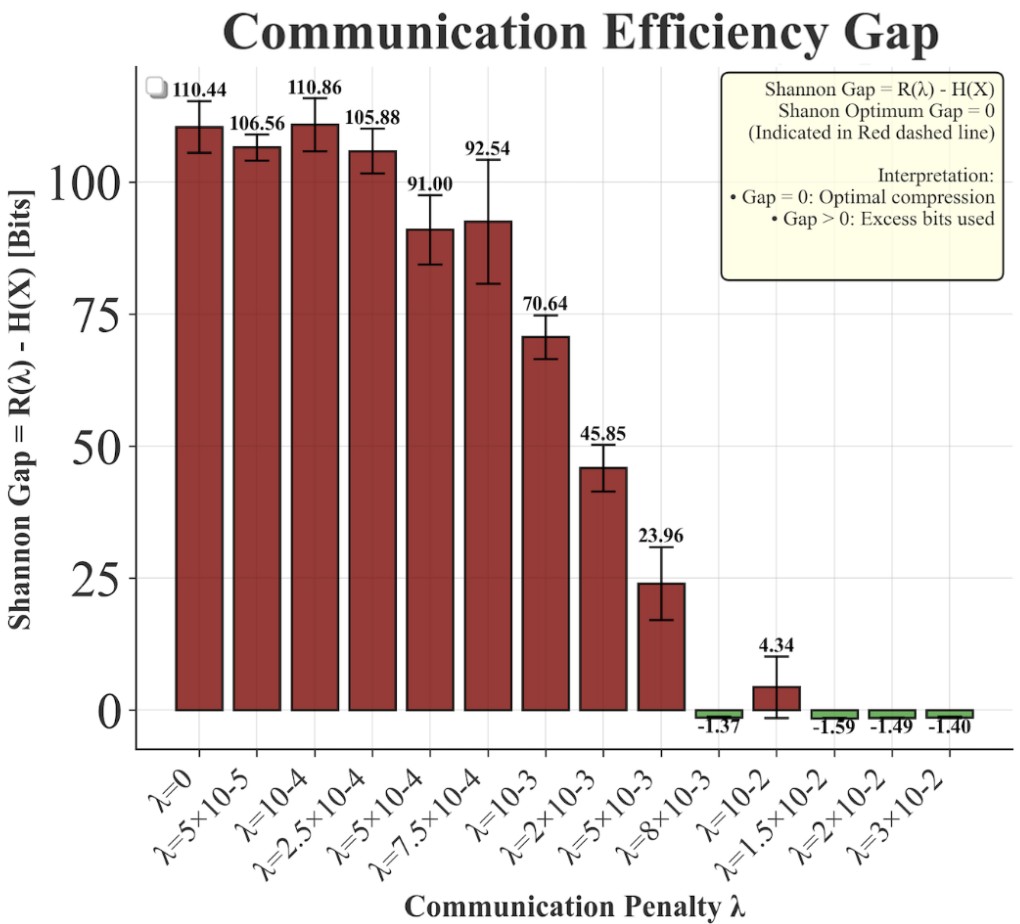

Figure 6: The Shannon Gap $(R(\lambda) - H(X))$ quantifies the communication inefficiency. The gap is large for small penalties but narrows significantly as the agent is forced to become more efficient.

reward magnitude increasing with the number of predators simultaneously on that cell. An episode is counted as a success if all predators capture the prey at the same time.

The **Traffic Junction (TJ)** environment is a cooperative, partially observable multi-agent grid world designed to simulate traffic coordination and collision avoidance. The environment is configured on a grid of a given dimension, with the complexity determined by the difficulty setting ('Medium' for 10 agents or 'Hard' for 20 agents), which defines the number of intersecting roads and possible paths. Agents, representing cars, are added stochastically into the system at entry points of the grid and are each assigned a predefined route to a destination on an opposite side of the junction. The task is partially observable, as each agent's observation is limited to a local 'vision' grid around its current position. An agent's observation at each timestep is a tuple containing its previous action, its assigned route ID, and a one-hot encoded representation of its local view that identifies roads and other cars. The discrete action space for each agent consists of two actions: *GAS* (move one step forward along the assigned path) or *BRAKE* (remain in the current position). The reward function encourages both safety and efficiency; agents receive a small penalty at each timestep ('-0.01') and a large penalty for collisions ('-10'), which occur if two or more agents occupy the same grid cell. An episode concludes when all active cars have reached their destinations, with the primary performance metric being the success rate, defined as completing an episode with zero collisions.

The **Google Research Football (GRF)** experiments utilize two distinct, challenging scenarios. The first, **GRF 3v1**, is a cooperative attacking drill where three player-controlled agents must coordinate their actions to score a goal against a single opponent goalkeeper. The second scenario,

**academy_corner**, is a set-piece task where the three agents must execute a corner kick to score. In both setups, each of the three agents uses the 'simple115v2' observation space, which provides a 115-dimensional feature vector containing information about player and ball positions, velocities, possession, and game mode. To encourage goal-scoring behavior, both scenarios use the sparse 'scoring' reward function, where the team receives a large positive reward for scoring a goal and a negative reward if the opponent scores.

We deliberately exclude benchmarks like the StarCraft Multi-Agent Challenge (SMAC (Samvelyan et al., 2019) and SMACv2 (Ellis et al., 2023)) from our evaluation. While they are standard in MARL, their design often makes it difficult to isolate the specific contribution of a learned communication protocol. As demonstrated in (de Witt et al., 2020), many scenarios in these environments can be solved effectively by non-communicating independent learners (e.g., IPPO) that coordinate implicitly by observing the shared environment state. This principle largely extends to SMACv2; because agents can infer team strategy and enemy positions from their local observations, the necessity for explicit, learned communication is often diminished. To properly evaluate a communication framework, it is essential to use environments where informational asymmetries create a clear and undeniable bottleneck, making successful coordination impossible or highly suboptimal without explicit information sharing. Our chosen environments—Traffic Junction, Predator-Prey, and Google Research Football—are all designed with such bottlenecks at their core.

Furthermore, we intentionally employ sparse reward functions across our chosen benchmarks, where agents receive an informative signal primarily upon reaching a terminal state (i.e., task success or failure). This design choice is critical for isolating the performance of the communication protocol itself. Using dense, intermediate rewards would introduce a confounding factor: the temporal credit assignment problem. It would become ambiguous whether an agent's success is due to learning an effective communication strategy or simply learning to exploit the local, intermediate reward signals. By tying the primary reward to the final outcome, the overall success of a trial becomes a direct measure of the team's ability to coordinate effectively over the entire episode. In our partially observable settings, this coordination is fundamentally dependent on the quality of the information transmitted, allowing us to draw clearer conclusions about how well a communication framework facilitates collective success.

# F  QUANTIZATION WITH STRAIGHT THROUGH ESTIMATOR.

---
**Algorithm 1** Fake Quantization

---
1: **function** APPLYFAKEQUANTIZATION(tensor, B)
2:                                                                    ▷ Setup quantization range for B bits (e.g., B=8)
3:     $q_{\min} \leftarrow 0$
4:     $q_{\max} \leftarrow 2^B - 1$
5:                                                                    ▷ Calculate min/max over the entire tensor
6:     $min\_val \leftarrow \min(tensor)$
7:     $max\_val \leftarrow \max(tensor)$
8:     **if** $min\_val = max\_val$ **then**                    ▷ Avoid division by zero in scale calculation
9:         $min\_val \leftarrow min\_val - 0.01$
10:         $max\_val \leftarrow max\_val + 0.01$
11:     **end if**
12:                                                                    ▷ Calculate quantization parameters
13:     $scale \leftarrow (max\_val - min\_val)/(q_{\max} - q_{\min})$
14:     $zero\_point \leftarrow \text{round}(q_{\min} - min\_val/scale)$
15:                                                        ▷ Simulate quantization and dequantization for gradient flow
16:     $scaled \leftarrow tensor/scale + zero\_point$
17:     $rounded \leftarrow \text{round}(scaled)$
18:     $clamped \leftarrow \text{clamp}(rounded, q_{\min}, q_{\max})$
19:     $shifted \leftarrow clamped - zero\_point$
20:     $dequantized \leftarrow shifted \times scale$
21:     **return** $dequantized$
22: **end function**

---

# G  DETAIL ANALYSIS OF DDCL AGAINST 32-FP AND QUANTIZED STE

Table 1: Efficiency Analysis for Traffic Junction (Medium) with 95% Confidence Intervals. DDCL variants are compared against their original 32-bit and 8-bit STE baselines.

| Algorithm | Comparison | Success Gain (%) | Comms Saved (OOM) | Success Efficiency |
|---|---|---|---|---|
| MAGIC | vs. Original | 0.84 (-5.74, 11.36) | 1.12 (1.07, 1.18) | 0.75 (-5.04, 10.23) |
| | vs. STE_8 | 43.25 (1.83, 164.40) | 0.52 (0.47, 0.58) | 83.07 (3.20, 314.28) |
| GAComm | vs. Original | 3.73 (-4.74, 14.18) | 1.00 (0.99, 1.01) | 3.73 (-4.73, 14.24) |
| | vs. STE_8 | 8.76 (-0.24, 17.51) | 1.05 (1.03, 1.06) | 8.37 (-0.23, 16.75) |
| IC3Net | vs. Original | -4.36 (-37.96, 19.40) | 1.54 (1.52, 1.55) | -2.84 (-24.59, 12.64) |
| | vs. STE_8 | 54.24 (-5.76, 156.33) | 0.26 (0.24, 0.27) | 211.70 (-22.45, 622.84) |
| TarMAC | vs. Original | 27.73 (-3.47, 76.54) | 1.03 (1.01, 1.05) | 26.93 (-3.30, 74.18) |
| | vs. STE_8 | 32.89 (10.86, 57.77) | 2.26 (2.24, 2.27) | 14.58 (4.79, 25.65) |
| MAPPO | vs. Original | 0.04 (-0.15, 0.24) | 4.66 (4.66, 4.66) | 0.01 (-0.03, 0.05) |
| | vs. STE_8 | -0.17 (-0.33, 0.01) | 4.06 (4.06, 4.06) | -0.04 (-0.08, 0.00) |

When compared to the original 32-bit float baselines in the Traffic Junction (Medium) (table 1) environment, the DDCL variants achieve substantial communication savings without a significant change in task performance. Communication is reduced by at least an order of magnitude (OOM) for all architectures, ranging from a 10x reduction for GAComm (1.00 OOM) to an over 45,000x reduction for MAPPO (4.66 OOM). While the mean success rates fluctuate—from a 27.7% gain for TarMAC to a 4.4% drop for IC3Net—none of these changes are statistically significant, as all 95% confidence intervals contain zero. In contrast, DDCL demonstrates a clear and often statistically significant performance advantage over naive 8-bit fixed quantization (STE). The DDCL variants of TarMAC and MAGIC achieve significant success rate gains of 32.9% and 43.3% respectively over their STE8 counterparts, with confidence intervals entirely above zero. This shows that DDCL provides a robust method for achieving massive, near-lossless compression that is superior to a fixed low-precision approach.

In the complex Traffic Junction (Hard) (table 2) environment, the results reveal statistically significant and highly varied trade-offs when applying DDCL. For IC3Net, DDCL provides a clear and statistically significant 'win-win,' improving the success rate by 6.0% (95% CI: [0.05, 13.35]) while reducing communication by over 30x (1.57 OOM) compared to the original baseline. In contrast, TarMAC exhibits a significant performance-for-efficiency trade-off, with its success rate dropping by a statistically significant 5.8% (95% CI: [-10.68, -0.57]) in exchange for a 10x communication saving. MAPPO showcases the most extreme near-lossless compression, cutting communication by over five orders of magnitude (5.26 OOM) with no statistically significant impact on its success rate. The comparison against fixed-quantization STE baselines is also nuanced; while DDCL significantly outperforms some STE variants (e.g., GAComm vs. STE4), it is also significantly outperformed by others (e.g., TarMAC vs. STE4), highlighting a complex interaction between adaptive precision and different architectures in this challenging scenario.

In the Predator-Prey (Medium) (table 3) environment, the DDCL variants generally maintain the performance of the original high-precision models while significantly reducing communication, though none of the success rate changes are statistically significant. For instance, MAPPO shows a large mean success gain of 27.5%, but with high variance, while cutting communication by over three orders of magnitude (3.71 OOM). TarMAC and IC3Net show modest positive gains in success rate alongside a greater than 10x reduction in bandwidth. In contrast, DDCL demonstrates a clear and statistically significant performance advantage over naive 8-bit quantization (STE) for MAGIC, which improves its success rate by 7.5% (95% CI: [2.20, 12.98]).

In the complex Predator-Prey (Hard) (table 4) environment, DDCL's impact reveals clear, statistically significant trade-offs and advantages. When compared to its original high-precision baseline, MAPPO exhibits a statistically significant 9.7% drop in success rate (95% CI: [-11.73, -8.44]) in exchange for a massive 7,600x reduction in communication (3.88 OOM). While other DDCL variants like TarMAC show a large mean improvement in success rate (+49.9%), high variance renders

Table 2: Efficiency Analysis for Traffic Junction (Hard) with 95% Confidence Intervals.

| Algorithm | Comparison | Success Gain (%) | Comms Saved (OOM) |
|---|---|---|---|
| MAGIC | vs. Original | 0.81 (-9.81, 12.07) | 1.29 (1.21, 1.41) |
| | vs. STE_4 | 10.27 (-2.11, 23.62) | 0.31 (0.30, 0.32) |
| | vs. STE_8 | 8.44 (-1.68, 19.36) | 0.61 (0.60, 0.63) |
| | vs. STE_16 | 10.05 (-0.81, 21.57) | 0.92 (0.90, 0.93) |
| GAComm | vs. Original | -10.05 (-17.90, 0.33) | 1.02 (1.00, 1.05) |
| | vs. STE_4 | 10.10 (2.54, 20.54) | 0.81 (0.78, 0.82) |
| | vs. STE_8 | 1.12 (-3.35, 5.69) | 1.06 (1.04, 1.08) |
| | vs. STE_16 | 2.97 (-1.63, 7.59) | 1.41 (1.39, 1.43) |
| IC3Net | vs. Original | 5.98 (0.05, 13.35) | 1.57 (1.40, 1.67) |
| | vs. STE_4 | 5.58 (-1.58, 13.10) | 1.13 (0.96, 1.24) |
| | vs. STE_8 | 6.75 (1.10, 14.23) | 0.24 (0.06, 0.34) |
| | vs. STE_16 | 2.47 (-2.57, 8.66) | 1.74 (1.57, 1.84) |
| TarMAC | vs. Original | -5.79 (-10.68, -0.57) | 1.09 (1.07, 1.11) |
| | vs. STE_4 | -6.87 (-11.12, -2.06) | 0.88 (0.86, 0.90) |
| | vs. STE_8 | -4.16 (-9.77, 1.67) | 2.32 (2.30, 2.34) |
| | vs. STE_16 | -6.27 (-10.90, -1.35) | 1.49 (1.46, 1.50) |
| MAPPO | vs. Original | 0.04 (-0.15, 0.22) | 5.26 (5.26, 5.26) |
| | vs. STE_4 | -0.01 (-0.17, 0.15) | 4.66 (4.66, 4.66) |
| | vs. STE_8 | -0.05 (-0.21, 0.12) | 4.66 (4.66, 4.66) |
| | vs. STE_16 | -0.00 (-0.17, 0.16) | 4.66 (4.66, 4.66) |

Table 3: Efficiency Analysis for Predator-Prey (Medium) with 95% Confidence Intervals.

| Algorithm | Comparison | Success Gain (%) | Comms Saved (OOM) |
|---|---|---|---|
| MAGIC | vs. Original | -2.38 (-6.87, 1.47) | 1.02 (0.93, 1.12) |
| | vs. STE_8 | 7.51 (2.20, 12.98) | 0.26 (0.20, 0.33) |
| GAComm | vs. Original | -2.50 (-7.50, 2.62) | 1.00 (0.93, 1.06) |
| | vs. STE_8 | 2.85 (-1.52, 6.05) | 0.71 (0.64, 0.76) |
| IC3Net | vs. Original | 0.40 (-1.85, 2.57) | 1.56 (1.32, 1.83) |
| | vs. STE_8 | 1.12 (-1.31, 3.29) | 0.79 (0.56, 1.06) |
| TarMAC | vs. Original | 3.35 (-2.16, 9.80) | 1.17 (0.98, 1.44) |
| | vs. STE_8 | 3.81 (-0.81, 9.28) | 2.57 (2.39, 2.84) |
| MAPPO | vs. Original | 27.53 (-7.53, 136.43) | 3.71 (3.56, 3.91) |
| | vs. STE_8 | -1.34 (-4.35, 1.85) | 3.31 (3.30, 3.32) |

these changes statistically insignificant. The most compelling results emerge in comparison to fixed-quantization (STE) baselines, where DDCL's adaptive approach proves vastly superior for several architectures. For MAGIC, DDCL achieves statistically significant success rate gains of 155.6%, 109.8%, and 128.8% over the STE4, STE8, and STE16 variants, respectively. Similarly, GAComm also shows significant improvements of 17.8% and 75.9% over its STE4 and STE8 counterparts.

In the highly complex and sparse-reward Google Research Football (GRF) 3v1 (table 5) environment, the DDCL variant of IC3Net achieves a transformative and statistically significant performance breakthrough. Compared to its original high-precision baseline, it increases the success rate by a remarkable 467.1% (95% CI: [12.34, 428.32]) while simultaneously reducing communication by over an order of magnitude (1.67 OOM). While other architectures like MAGIC (+176.3%) and TarMAC (+37.1%) also show large mean performance gains, the extremely high variance in this environment renders these changes statistically insignificant. When compared to fixed-quantization baselines, DDCL-IC3Net again demonstrates a statistically significant advantage over its 4-bit STE counterpart with a 225.4% increase in success rate. For other architectures, the high variance leads

Table 4: Efficiency Analysis for Predator-Prey (Hard) with 95% Confidence Intervals.

| Algorithm | Comparison | Success Gain (%) | Comms Saved (OOM) |
|---|---|---|---|
| MAGIC | vs. Original | 13.75 (-7.52, 47.80) | 0.98 (0.93, 1.04) |
| | vs. STE_4 | 155.62 (98.73, 227.81) | -0.03 (-0.07, 0.01) |
| | vs. STE_8 | 109.78 (65.11, 164.39) | 0.27 (0.24, 0.32) |
| | vs. STE_16 | 128.76 (73.31, 211.74) | 0.57 (0.53, 0.62) |
| GAComm | vs. Original | -0.84 (-19.26, 27.17) | 1.11 (1.01, 1.20) |
| | vs. STE_4 | 17.81 (2.64, 31.62) | 0.40 (0.31, 0.46) |
| | vs. STE_8 | 75.90 (41.16, 128.53) | 1.03 (0.85, 1.17) |
| | vs. STE_16 | 12.42 (-2.52, 26.27) | 1.00 (0.93, 1.07) |
| IC3Net | vs. Original | 4.21 (-17.23, 28.81) | 1.40 (1.20, 1.60) |
| | vs. STE_4 | 37.09 (-16.84, 236.49) | 0.44 (0.09, 0.67) |
| | vs. STE_8 | -16.31 (-31.33, 0.77) | 0.39 (0.27, 0.57) |
| | vs. STE_16 | 1.65 (-16.98, 18.95) | 0.94 (0.74, 1.16) |
| TarMAC | vs. Original | 49.87 (-3.18, 98.50) | 1.24 (1.00, 1.52) |
| | vs. STE_4 | -7.20 (-47.14, 35.30) | 0.29 (0.10, 0.55) |
| | vs. STE_8 | -29.19 (-57.92, 0.85) | 2.61 (2.42, 2.87) |
| | vs. STE_16 | -5.25 (-43.76, 27.72) | 0.90 (0.70, 1.17) |
| MAPPO | vs. Original | -9.74 (-11.73, -8.44) | 3.88 (3.87, 3.90) |
| | vs. STE_4 | 4.34 (-4.92, 22.61) | 3.67 (3.66, 3.68) |
| | vs. STE_8 | 0.62 (-4.87, 8.50) | 3.67 (3.66, 3.68) |
| | vs. STE_16 | 7.83 (-1.36, 24.43) | 3.67 (3.66, 3.68) |

Table 5: Efficiency Analysis for GRF (3 vs 1) with 95% Confidence Intervals. Note the high variance across many runs.

| Algorithm | Comparison | Success Gain (%) | Comms Saved (OOM) |
|---|---|---|---|
| MAGIC | vs. Original | 176.31 (-69.47, 750.78) | 1.21 (1.01, 1.48) |
| | vs. STE_4 | -23.48 (-92.93, 180.15) | 0.33 (0.15, 0.56) |
| | vs. STE_8 | -16.92 (-90.29, 111.47) | 0.59 (0.41, 0.84) |
| | vs. STE_16 | -29.33 (-91.96, 89.51) | 0.88 (0.71, 1.12) |
| GAComm | vs. Original | 0.72 (-42.79, 74.34) | 1.06 (0.96, 1.15) |
| | vs. STE_4 | 261.76 (-4.71, 1077.58) | 0.55 (0.36, 0.78) |
| | vs. STE_8 | 54.95 (-23.43, 234.09) | 0.69 (0.49, 0.85) |
| | vs. STE_16 | 7.57 (-42.65, 111.40) | 1.03 (0.92, 1.14) |
| IC3Net | vs. Original | 467.14 (12.34, 428.32) | 1.67 (1.27, 2.10) |
| | vs. STE_4 | 225.40 (33.27, 554.78) | 0.82 (0.48, 1.22) |
| | vs. STE_8 | 1292.67 (-37.40, 14809.12) | 0.81 (0.51, 1.23) |
| | vs. STE_16 | 79.26 (-45.34, 297.04) | 1.55 (1.20, 1.97) |
| TarMAC | vs. Original | 37.08 (-81.53, 350.92) | 0.97 (0.80, 1.19) |
| | vs. STE_4 | 89.83 (-75.69, 537.55) | 0.10 (-0.06, 0.31) |
| | vs. STE_8 | -14.29 (-83.67, 107.71) | 2.44 (2.29, 2.66) |
| | vs. STE_16 | -12.97 (-85.97, 141.52) | 0.68 (0.54, 0.88) |
| MAPPO | vs. Original | -22.62 (-50.12, 5.87) | 4.20 (4.12, 4.27) |
| | vs. STE_4 | 23.72 (-31.96, 121.77) | 3.42 (3.39, 3.44) |
| | vs. STE_8 | 26.05 (-18.70, 72.32) | 3.42 (3.39, 3.44) |
| | vs. STE_16 | 32.98 (-16.41, 89.32) | 3.42 (3.39, 3.44) |

to non-significant differences against STE, highlighting the stochastic nature of learning in this challenging domain.

Table 6: Efficiency Analysis for GRF (Corner) with 95% Confidence Intervals. Note the extremely high variance in this environment, leading to non-significant results.

| Algorithm | Comparison | Success Gain (%) | Comms Saved (OOM) |
|---|---|---|---|
| MAGIC | vs. Original | 261.51 (-45.28, 1247.05) | 1.17 (1.03, 1.34) |
| | vs. STE_8 | -9.85 (-80.17, 135.42) | 0.62 (0.48, 0.82) |
| GAComm | vs. Original | 15.80 (-68.64, 206.74) | 1.18 (1.05, 1.31) |
| | vs. STE_8 | -16.37 (-78.54, 107.53) | 0.69 (0.49, 0.94) |
| IC3Net | vs. Original | 59.51 (-57.08, 383.73) | 1.00 (0.63, 1.31) |
| | vs. STE_8 | 69.22 (-43.02, 308.17) | 0.31 (0.08, 0.59) |
| TarMAC | vs. Original | 40.13 (-40.65, 135.67) | 1.00 (0.87, 1.19) |
| | vs. STE_8 | -25.91 (-72.42, 51.12) | 2.49 (2.36, 2.68) |
| MAPPO | vs. Original | -3.35 (-46.75, 47.12) | 4.47 (4.40, 4.54) |
| | vs. STE_8 | 47.44 (-34.34, 214.70) | 3.91 (3.79, 4.01) |

In the highly stochastic Google Research Football (GRF) Corner (table 6) environment, the experimental results are characterized by extremely high variance, preventing definitive conclusions on performance changes. While most DDCL variants show a strong positive trend in mean success rate compared to their original baselines—most notably MAGIC with a +261.5% mean gain—the wide 95% confidence intervals for all architectures span zero, rendering these changes not statistically significant. The one consistent result is a significant reduction in communication cost, with MAPPO achieving the highest compression by over four orders of magnitude (4.47 OOM). Similarly, comparisons against the 8-bit STE baseline are also not statistically significant due to high variance. This suggests that while DDCL consistently enables drastic communication savings, the complex and sparse-reward nature of this task leads to a highly variable impact on final task performance.

## H    EPISODIC REWARDS VS. COMMUNICATION BITS

This section provides supplementary results that complement the success rate analysis in the main paper. While success rate captures task completion, the cumulative episodic reward offers a finer-grained measure of performance, often reflecting the efficiency and speed with which agents solve the task. The following plots show the Pareto frontiers for episodic reward versus communication bits across the Traffic Junction and Predator-Prey benchmarks.

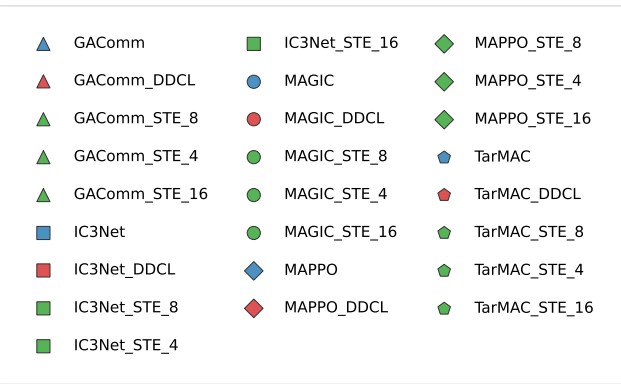

Figure 7: Shared legend for episodic reward vs communication bandwidth experimental results figures.

**Analysis of Results.**    The results for episodic reward, shown in fig. 8, reinforce the conclusions from the success rate analysis in the main paper. Across all three benchmarks—Traffic Junction, Predator-Prey, and Google Research Football—the DDCL-enhanced agents consistently demon-

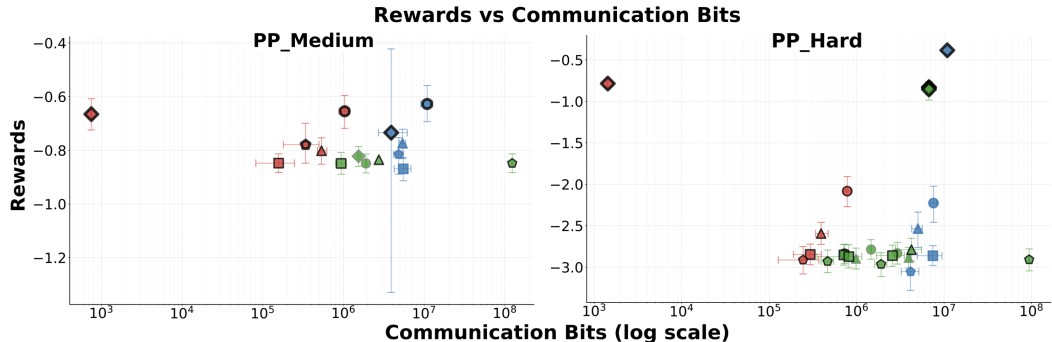

(a) In both Medium (left) and Hard (right) settings, the DDCL variants (red) consistently achieve similar or higher rewards than their original counterparts while operating at a fraction of the communication cost.

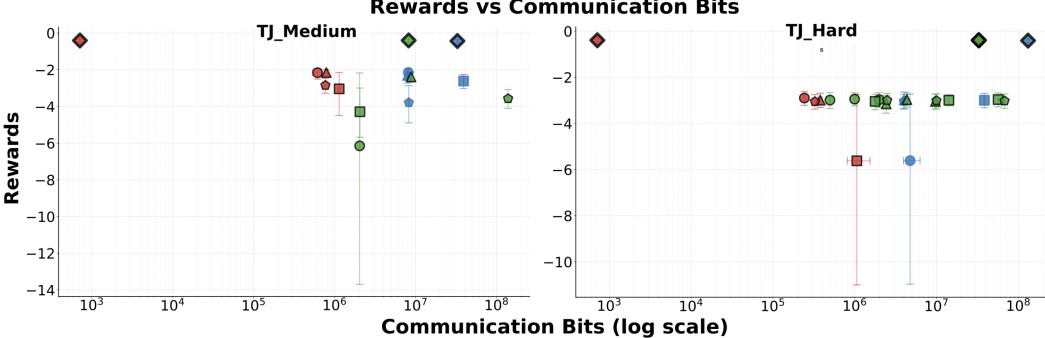

(b) The DDCL agents again establish the Pareto frontier. In both the Medium and Hard setting, the DDCL variants achieve nearly similar rewards in comparison to their baselines while requiring orders of magnitude less communication bits, indicating more efficient solutions.

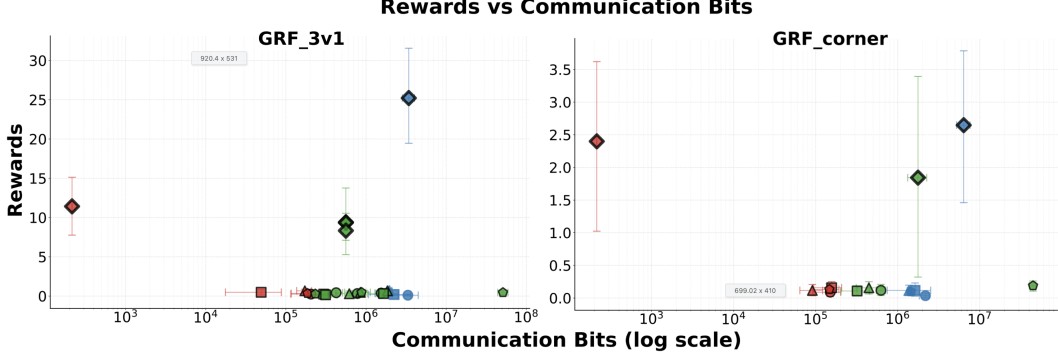

(c) In both the 3v1 (left) and Corner (right) scenarios, the DDCL variants dominate the left quadrant, achieving near similar rewards as the original algorithms but with the lowest communication cost. This demonstrates that the efficiency gains do not compromise the ability to find similar capable policies in complex, sparse-reward tasks.

Figure 8: **Episodic reward versus communication bandwidth across all benchmark environments.** Each point represents an algorithm variant's mean performance (Episodic Reward) and communication cost (Bits per episode, log scale) over 5 different seeds. Error bars denote 95% confidence intervals. The top-left of each plot represents the ideal outcome (high episodic reward, low communication cost), while the bottom-right is the worst. Our DDCL-enhanced variants (red markers) consistently operate on the left side of the plots, demonstrating significant communication savings while maintaining or gaining more episodic rewards. The global Pareto frontier, representing the best possible trade-offs, is marked with a **thick black border**, while algorithm-specific frontiers are marked with a **thin black border**.

strate a superior trade-off between performance and communication cost. In every scenario, the DDCL variants (red markers) form the Pareto-optimal frontier, achieving the highest episodic rewards for a given communication budget. For example, in Traffic Junction Hard (fig. 8b), top-performing DDCL agents achieve rewards near -2, while baseline agents using two to four orders of magnitude more communication are clustered at rewards between -10 and -20. This pattern confirms that the communication efficiency gained from DDCL does not come at the cost of solution quality; on the contrary, it enables agents to learn more effective policies that solve tasks more quickly and with fewer penalties, thereby accumulating higher rewards.

### H.1 SENSITIVITY TO HYPERPARAMETERS

**Effect of Communication Cost** ($\lambda$)**.** Across all DDCL-based experiments, we vary the communication cost coefficient $\lambda$ over several orders of magnitude. This allows us to trace the Pareto frontier for each algorithm, explicitly showing the trade-off between task performance and communication bandwidth. This demonstrates that $\lambda$ provides a direct and effective control over the desired operating point on this curve.

**Effect of Quantization Granularity** ($\delta$)**.** We perform an ablation study on the width of the uniform noise interval, $\delta$. This parameter directly controls the coarseness of the quantization grid. We investigate how performance and bandwidth are affected by finer ($\delta \to 0$) versus coarser (larger $\delta$) quantization, testing the robustness of our framework to this key hyperparameter.

## I SENSITIVITY TO HYPERPARAMETERS AND ADDITIONAL RESULTS

This section provides a detailed analysis of the sensitivity of our DDCL framework to its two primary hyperparameters: the communication cost coefficient $\lambda$ and the quantization granularity $\delta$. We also provide supplementary reward-based performance plots and a legend for all experimental figures.

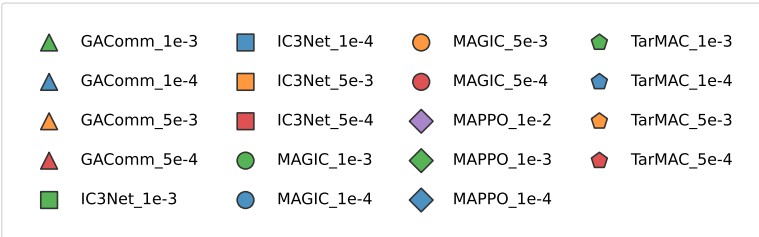

Figure 9: Shared legend for $\lambda$ experimental results figures. Marker shapes denote the baseline algorithm (e.g., Circle for TarMAC, Square for IC3Net), while colors denote the algorithm family (e.g., Blue for $\lambda = 1e - 4$, Green for $\lambda = 1e - 3$ etc.).

**Effect of Communication Cost** ($\lambda$)**.** Theoretically, the communication cost coefficient, $\lambda$, should act as a **regularization parameter** that explicitly controls the trade-off between maximizing task reward and minimizing communication cost. A small $\lambda$ should prioritize performance, while a large $\lambda$ should prioritize efficiency. Our empirical results, shown in fig. 10, validate this expectation perfectly. Increasing the value of $\lambda$ consistently and smoothly moves the agents from high-performance, high-cost to low-cost, lower-performance region of the plots, effectively tracing out

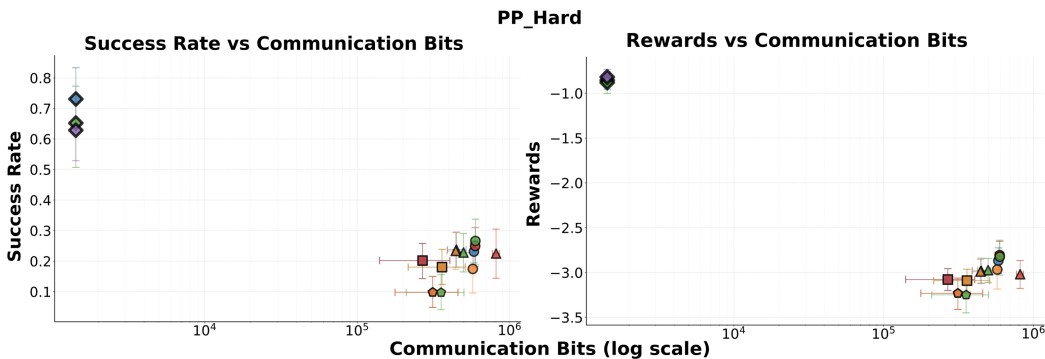

(a) This plot evaluates the framework's sensitivity to the communication coefficient, $\lambda$, revealing a high degree of robustness. For each algorithm, the performance outcomes for different $\lambda$ values are statistically similar except for MAPPO with slight variation observed, indicating that the framework mostly does not require extensive tuning of this hyperparameter.

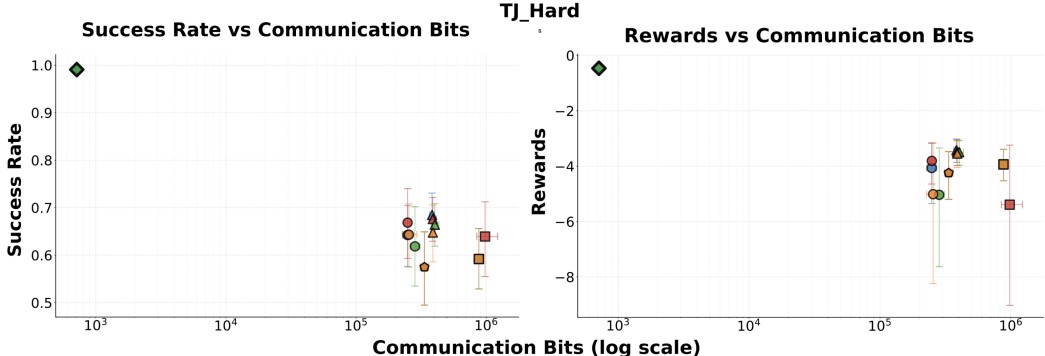

(b) This plot evaluates sensitivity to the communication coefficient, $\lambda$, in the Traffic Junction environment, revealing that the optimal choice is not architecture-dependent.

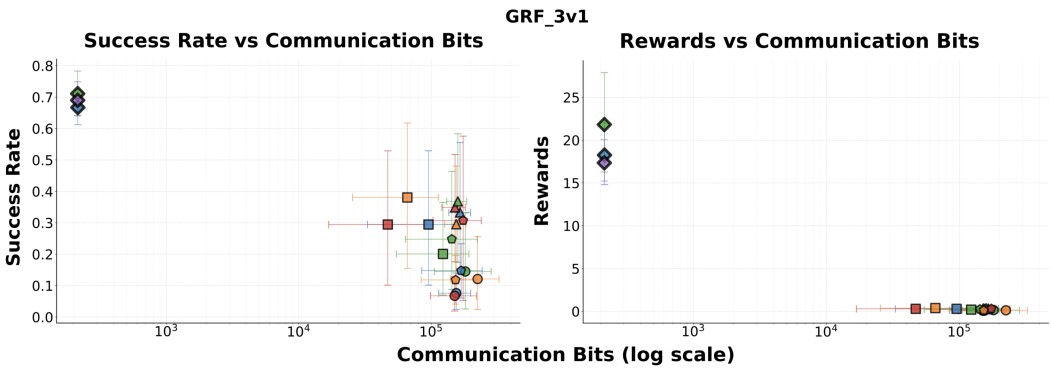

(c) This plot evaluates sensitivity to the communication coefficient, $\lambda$, in the GRF environment, again showing that the optimal choice is not architecture-dependent.

Figure 10: **Sensitivity analysis for the communication cost coefficient, $\lambda$, across three benchmark environments:** (a) Predator-Prey Hard, (b) Traffic Junction Hard, and (c) GRF 3v1. Each plot shows the trade-off between Success Rate and Episodic Rewards against Communication Bits (log scale), with points representing the mean over multiple seeds and error bars denoting 95% confidence intervals. The results demonstrate a consistent and predictable trend across all environments and architectures. Crucially, performance is often robust across a range of $\lambda$ values, suggesting that the framework does not require extensive hyperparameter tuning to find an effective point on the performance-efficiency spectrum.

the Rate-Distortion curve for each architecture. Crucially, our results also reveal that performance is often robust across a range of intermediate $\lambda$ values, as seen in the Predator-Prey environment. This indicates that while even slight variations in $\lambda$ predictably affect the communication rate, the framework is not overly sensitive and does not require extensive tuning to find a desirable operating point that balances high performance with significant communication savings.

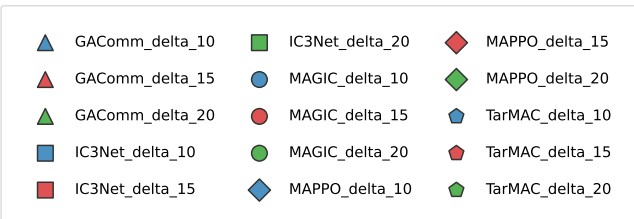

Figure 11: Shared legend for $\delta$ experimental results figures. Marker shapes denote the baseline algorithm (e.g., Circle for TarMAC, Square for IC3Net), while colors denote the algorithm family (e.g., Red for $\delta = 15$, Blue for $\delta = 10$ and Green for $\delta = 20$).

**Effect of Quantization Granularity ($\delta$).** Ideally, the quantization granularity, $\delta$, should strike a balance between **expressivity** and **learning stability**. A grid that is too fine (small $\delta$) offers high precision but can complicate the exploration problem, while a grid that is too coarse (large $\delta$) is easier to learn but may create an information bottleneck that limits peak performance. Our empirical results validate this expectation, revealing that the optimal choice of $\delta$ is highly dependent on the task and agent architecture, as shown in the detailed plots fig. 12. In a relatively simple task like Predator-Prey, performance is robust across a range of $\delta$ values, indicating a wide "sweet spot." However, in more complex environments like Traffic Junction and GRF, we observe that specific architectures achieve a superior performance-efficiency trade-off at a particular intermediate granularity ($\delta = 15$), while a finer grid ($\delta = 10$) can sometimes yield higher absolute success at the cost of increased communication. This confirms that even slight variations in $\delta$ can meaningfully affect the outcome, and its optimal value depends on the specific informational requirements of the task at hand.

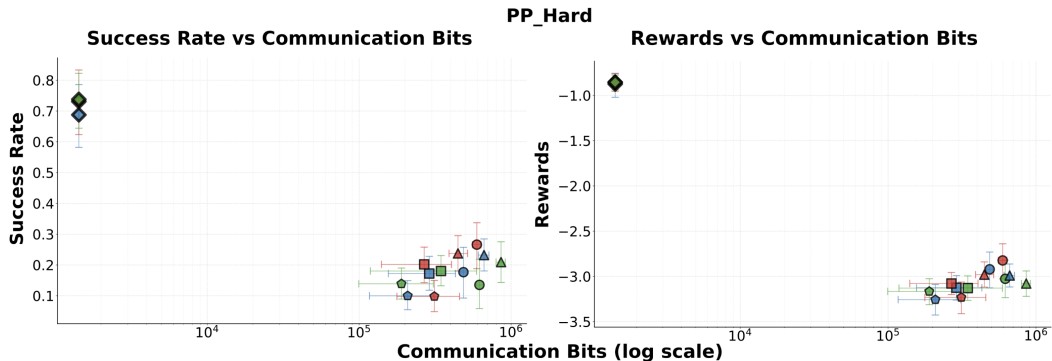

(a) This plot evaluates the framework's sensitivity to the quantization granularity, $\delta$, revealing a high degree of robustness. For each algorithm, the performance outcomes for different $\delta$ values are statistically similar, indicating that the framework does not require extensive tuning of this hyperparameter.

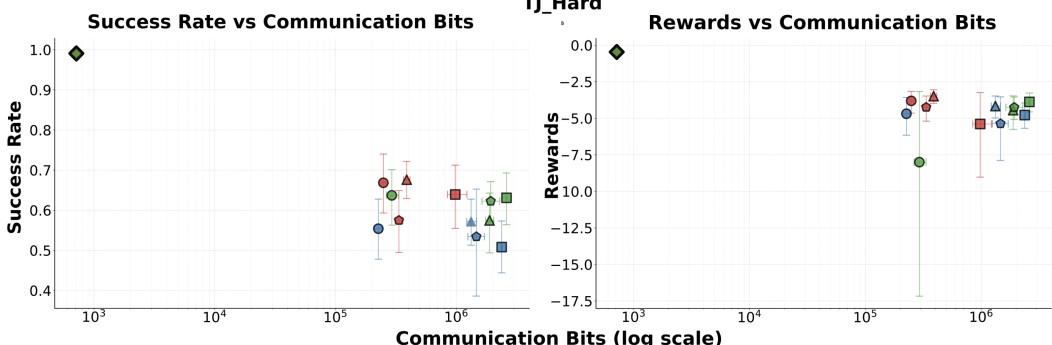

(b) This plot evaluates sensitivity to the quantization granularity, $\delta$, in the Traffic Junction environment, revealing that the optimal choice is architecture-dependent. While MAPPO and MAGIC show robust performance across different $\delta$ values, other architectures like IC3Net, GAComm, and TarMAC achieve a superior performance trade-off specifically with an intermediate granularity ($\delta = 15$).

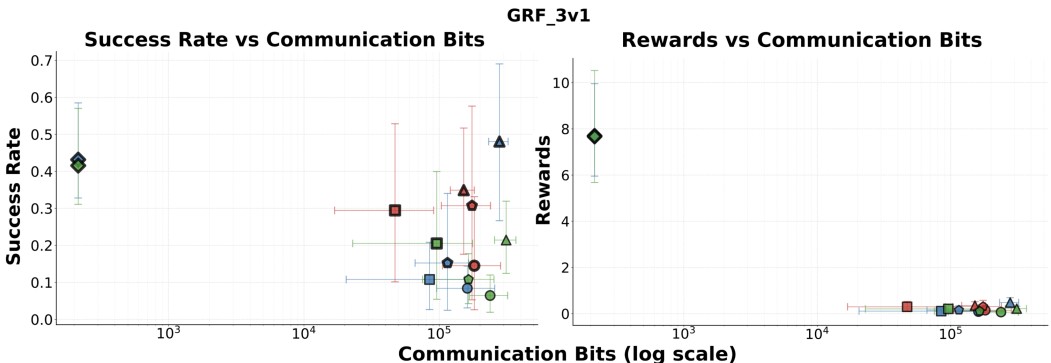

(c) This plot evaluates sensitivity to the quantization granularity, $\delta$, in the GRF environment, again showing that the optimal choice is architecture-dependent. While MAPPO's performance is robust across the tested values, IC3Net and TarMAC achieve their best performance-efficiency trade-offs with an intermediate granularity of $\delta = 15$. GAComm also finds $\delta = 15$ to be most efficient, though a finer granularity of $\delta = 10$ can yield a higher absolute success rate for an increased communication cost.

Figure 12: This figure evaluates the framework's sensitivity to the quantization granularity, $\delta$, across all benchmarks, revealing that the optimal choice is highly dependent on both the environment and the agent architecture. While performance is robust to $\delta$ in some tasks (e.g., Predator-Prey), other environments show that specific architectures achieve a superior performance-efficiency trade-off with an intermediate granularity ($\delta = 15$), as seen with IC3Net and TarMAC in Traffic Junction and GRF. This highlights a complex interplay where no single $\delta$ value is universally optimal, and the ideal granularity must be considered in the context of the specific task and model being used.

