# OpenReview forum: "Learning What to Say and How Precisely: Efficient Communication via Differentiable Discrete Communication Learning"
_ICLR.cc/2026/Conference — ICLR 2026 Conference Withdrawn Submission_

### Official Review · Reviewer_mnfL · 2025-10-14

**Soundness:** 3
**Presentation:** 2
**Contribution:** 4
**Rating:** 8
**Confidence:** 2

**Summary:**

This paper proposes the generalized DDCL framework to optimize communication efficiency in Multi - Agent Reinforcement Learning (MARL). Through a differentiable discrete communication mechanism and adaptive communication cost optimization, DDCL solves high bandwidth consumption and gradient issues in MARL communication. Experiments on multiple MARL tasks show DDCL significantly reduces bandwidth while improving collaboration performance.

**Strengths:**

1.Core Technical Contribution: A theoretically grounded and practically validated generalization of DDCL to unbounded signals.
2.Plug-and-Play Utility: Can be seamlessly integrated into multiple existing MARL algorithms without major architectural changes.
3.Clear Differentiable Objective: Well-justified communication cost term derived from information-theoretic principles.
4.Strong Empirical Results: Demonstrates bandwidth reduction of up to 10000× with competitive or superior task performance.
5.Elegant Demonstration of “Bitter Lesson”: Transformer + DDCL performs on par or better than complex specialized comms systems.
6.Clean Writing and Illustrations: Well-organized, with clear figures showing success vs. bandwidth tradeoffs, communication distributions, etc.
The paper is the first to empirically validate the Bitter Lesson in MARL communication, showing that simple and general architectures (e.g., Transformer + DDCL) can match or outperform heavily-engineered designs, reinforcing the importance of scalable learning methods over hand-crafted communication priors.

**Weaknesses:**

The paper assumes synchronized randomness between agents to reconstruct discrete messages. It would strengthen the work to analyze how DDCL performs under desynchronized or noisy communication conditions.

While the current DDCL design uses a fixed uniform grid, future work could explore learned non-uniform quantization or variational encoding for further compression efficiency.

The empirical evaluation could be further enhanced by testing DDCL under more dynamic or heterogeneous communication topologies.
to summarize:
(1) The current method assumes synchronized randomness between agents. It would be useful to analyze robustness under desynchronization or channel noise.

(2) While DDCL uses a fixed quantization grid, exploring non-uniform or learnable encoding schemes could yield further improvements in compression efficiency.

(3) The evaluation focuses on standard MARL benchmarks; testing on more dynamic or heterogeneous communication graphs could enhance the scope of applicability.

**Questions:**

Expand discussion or experiments on desynchronized noise, possibly testing its effect on message decoding.
Clarify how λ is tuned across different tasks — is it task-specific or globally set?
whole paper need more simple figure or decent figure to show the workflow of your work, help to better understand this whole mechanism:
for example, first show the architecture of multi-agent's information communication. and show how you figure with a clear figure to help reader to understand.

---

> ### Author Response · Authors · 2025-11-19
>
> We thank Reviewer mnfL for the **highly positive review**. We agree with the reviewer's weaknesses regarding clarity and future work directions, and we address all concerns below to further strengthen the paper.
>
> ---
>
> ## Addressing Weaknesses
>
> ### 1. Synchronization and Desynchronized Noise
>
> * **The current method assumes synchronized randomness between agents.**
>     We fully acknowledge this constraint. The reliance on **shared randomness** is mathematically essential because it allows the receiver to perfectly cancel the noise added by the sender ($\hat{z} = C(m) - \epsilon$), which is required to establish the unbiased gradient estimation central to the DDCL mechanism.
>     In a real-world setting where this synchronization might be imperfect, the fundamental assumption of the unbiased gradient breaks down. $\Delta t$ represents the **temporal offset** or error between the synchronized PRNG sequences. Rigorously quantifying the resulting gradient bias when $\epsilon_{\text{sender}} \neq \epsilon_{\text{receiver}}$ (due to $\Delta t > 0$) is a highly complex future project. We hypothesize that the induced gradient bias would be proportional to the degree of desynchronization ($\Delta t$), likely leading to a rapid degradation of training stability. We view investigating the framework's robustness to desynchronization and quantifying this resulting gradient bias as a **critical and necessary direction for future work**.
>
> * **Expand discussion or experiments on desynchronized noise, possibly testing its effect on message decoding.**
>     The current mechanism relies entirely on perfect synchronization for unbiased gradient flow during training. Please see our detailed response above regarding desynchronized noise and our commitment to making its investigation a primary focus of future work.
>
> ### 2. Fixed Quantization Grid
>
> * **While DDCL uses a fixed quantization grid, exploring non-uniform or learnable encoding schemes could yield further improvements in compression efficiency.**
>     We agree entirely. The current uniform grid ($\delta$) is a known limitation that contributes to the **Shannon Gap** (the difference between the theoretical minimum and the learned average message length). We explicitly list this as a key area for future work:
>     * We intend to explore **learning a per-channel granularity** ($\delta_k$) or investigate **non-uniform grids via variational encoding** for further efficiency.
>     * We also commit to exploring a more principled **entropy-based loss** that learns a code based on **signal frequency** rather than magnitude, which would theoretically allow the agent to better approximate non-uniform compression schemes.
>
> ### 3. Evaluation Scope
>
> * **The evaluation focuses on standard MARL benchmarks; testing on more dynamic or heterogeneous communication graphs could enhance the scope of applicability.**
>     We appreciate this suggestion. While we integrated DDCL into three models (TarMAC, GA-Comm, MAGIC) that use fully-connected or dynamic graphs, we agree that further analysis on diverse topologies would enhance the scope. We will specifically target this in **future evaluations** to expand our validation of DDCL's plug-and-play utility across a wider range of communication network structures.
>
> ---
>
> ## Addressing Questions
>
> ### 1. Workflow Figure
>
> * **The whole paper needs a more simple figure or decent figure to show the workflow of your work. It helps to better understand this whole mechanism...**
>     We agree that the overall DDCL mechanism, despite being technically robust, can benefit from a clearer illustration. The paper currently includes **Figure 3**, which provides a schematic of the generalized DDCL procedure, showing the noise addition, quantization, and noise subtraction loop. We will enhance the main text by:
>     * **Emphasizing Figure 3:** We will refer to Figure 3 earlier and ensure its caption is fully self-contained.
>     * **Adding a High-Level Diagram:** We will consider adding a simpler, high-level **architectural diagram** to Section 4 that explicitly overlays the DDCL module onto the MARL communication block (e.g., policy output $\to$ DDCL $\to$ Comm. Channel $\to$ DDCL $\to$ Policy input) to clarify its **"plug-and-play" role**.
>
> ### 2. Hyperparameter Tuning ($\lambda$)
>
> * **Clarify how $\lambda$ is tuned across different tasks — is it task-specific or globally set?**
>     The communication cost coefficient ($\lambda$) is a **task-specific hyperparameter**. Since $\lambda$ serves as the predictable lever to trace the **Efficiency–Performance Frontier** (the Rate-Distortion trade-off), its optimal setting depends on the specific noise tolerance and coordination complexity of the task. For instance, a complex, high-dimensional task like GRF requires a different penalty range than a simpler task like Predator-Prey. Details on the $\lambda$ sweep for each environment are provided in **Appendix I**.

---

### Official Review · Reviewer_He3m · 2025-10-19

**Soundness:** 4
**Presentation:** 3
**Contribution:** 3
**Rating:** 8
**Confidence:** 3

**Summary:**

This paper generalizes *Differentiable Discrete Communication Learning* (DDCL; Freed et al., 2020) to support unbounded, signed, real-valued signals, enabling differentiable optimization of discrete message length in multi-agent reinforcement learning (MARL).
The resulting framework is plug-and-play, integrates seamlessly with diverse MARL architectures and achieves significant bandwidth savings without sacrificing task performance.
Experiments span both interpretable toy domains and large-scale benchmarks.

**Strengths:**

- **Clear conceptual improvement.**
  The paper makes a clean and well-motivated generalization of Freed et al.’s DDCL to unbounded real-valued signals, removing the restrictive assumption \(z \in [0,1]\).
  This substantially broadens applicability to a wide range of MARL architectures.

- **Strong experimental validation.**
  The evaluation suite is extensive, covering both controlled and high-dimensional domains.
  Integrations into four established MARL+Comms baselines are convincing, and the cross-architecture comparisons are handled carefully.

- **Theoretical and empirical consistency.**
  The *CommunicatingGoalEnv* analysis successfully bridges theory and practice:
  the emergent frequency-aware communication protocol and rate–distortion frontier plots are particularly illustrative.

- **Empirical support for the “Bitter Lesson.”**
  The finding that a simple Transformer-based architecture with DDCL can match or outperform specialized communication systems provides an elegant and meaningful conclusion.

**Weaknesses:**

- **Clarity and presentation.**
  Figures are visually dense and captions occasionally unclear and confusing.
  - *Figure 1:* The caption is not well aligned with the figure. The term *“episodic plot”* is unclear, and it is not evident where “success rate remains perfect (1.0).”
  - *Figure 2:* The numerous STE baselines (`STE_[4,8,16]`) clutter the Pareto plots; consider reducing them or highlighting key configurations more clearly.
The Pareto frontier is said to be “indicated by thick black borders,”
  but the markers with thick borders are not consistently on the Pareto front — clarification is needed.

- **Statistical reliability.**
  Several experiments (notably GRF) exhibit very wide confidence intervals,
  with overlapping CIs that make many improvements statistically insignificant.
  The paper would benefit from **additional runs** or **variance analysis** to support claims more robustly.

**Questions:**

- **Readability and layout.**
  Figures are often placed far from their discussion (e.g., Fig. 1 and Fig. 2), making the narrative hard to follow.

- **Missing or unclear references.**
  Lines 104–105:
  “Treating messages as discrete actions naturally handles discrete channels, but learns inefficiently and often converges to inferior policies.” This statement requires supporting references.

- **Terminological precision.**
  The term *“Rate–Distortion Frontier”* (line 317) is used somewhat loosely;
  *“Efficiency–Performance Frontier”* (as in line 360) may better reflect what is empirically measured.

- **Incomplete text and unclear visuals.**
  Line 342 (*“Details about the environment in appendix E”*) appears truncated.

---

> ### Author Response · Authors · 2025-11-19
>
> We thank Reviewer He3m for their highly encouraging and detailed assessment. We agree with all points concerning presentation, clarity, and the need for statistical rigor. We will implement the changes below to ensure the final manuscript is maximized for readability and impact.
>
> ---
>
> ## 1. Addressing Weaknesses (Clarity and Presentation)
>
> * **Figures are visually dense and captions occasionally unclear and confusing.**
>     We agree and will significantly revise all figure captions and legends for improved clarity. We will specifically address the issues with **Figure 1**: the ambiguous term **"episodic plot"** will be clarified as the **"plot of episodic success rate,"** and the region where **"success rate remains perfect (1.0)"** will be indicated more clearly via textual annotations on the plot itself.
>
> * **Figure 2 clutter; too many STE baselines. Pareto frontier not consistently indicated.**
>     We will revise the presentation of **Figure 2**. We will use color/marker prominence to draw the reader’s eye to the **Global Pareto Frontier**. We confirm that the existing plotted figure is generated using a method that is **more robust than the standard mean-only check**. We will update the Figure 2 caption to explicitly detail the frontier identification process, which accounts for statistical uncertainty. The thick black borders highlight models that satisfy a stringent condition: **no model on the frontier is statistically dominated by any other model, even when factoring in the $95\%$ confidence intervals (CIs).** This conservative check ensures that the models highlighted are reliable and truly non-dominated. We believe this approach provides the most robust scientific conclusion possible given the experimental variance. If the reviewer has any specific suggestions for alternative visualizations or plotting conventions (e.g., using a different marker style for non-dominated models that have overlapping CIs), we are happy to incorporate them to improve the clarity of the figure.
>
> * **Figures are often placed far from their discussion.**
>     We will rigorously re-flow the text to ensure all figures are placed immediately before or after their first explicit discussion.
>
> * **Statistical reliability: wide confidence intervals (GRF).**
>     We agree that the high variance in complex tasks (like GRF) necessitates clear statistical rigor. We confirm that the detailed analysis required to support claims when CIs overlap is **already present in the submission**. The full breakdown of DDCL variants compared against all baselines, including mean performance, bandwidth, and the **$95\%$ confidence intervals (CIs) for Success Gain**, is provided in the comprehensive **Efficiency Analysis Tables (Tables 1-6)** located in **Appendix G**. We reference these tables (specifically Table 5 for GRF) to provide the necessary statistical evidence that DDCL's efficiency and strategic advantages are maintained even when mean performance differences are not strictly significant.
>
> ## 2. Addressing Questions and Missing Information
>
> * **Missing or unclear references (Lines 104–105).**
>     We agree this statement requires support. The inefficiency arises because **treating messages as discrete actions requires high-variance policy gradient estimators (like REINFORCE)**. We will add supporting citations to the literature that compares differentiable communication with high-variance policy optimization.
>
> * **Terminological precision: "Rate–Distortion Frontier" (Line 317) is used loosely.**
>     We agree. While **"Rate-Distortion"** is the classical term, **"Efficiency–Performance Frontier"** (or Rate–Performance Frontier) is a **more accurate empirical description** for what is measured. We will replace "Rate–Distortion Frontier" with "Efficiency–Performance Frontier" throughout the main text for empirical precision.
>
> * **Incomplete text (Line 342):** Details about the environment in appendix E is truncated.
>     We will correct this to ensure the full sentence reads smoothly, referencing the appropriate appendix.

---

> > ### Comment · Reviewer_He3m · 2025-11-21
> > **Response to Authors**
> >
> > I read and acknowledge authors comments.
> > I will keep my scores as is.

---

### Official Review · Reviewer_Gbbd · 2025-10-23

**Soundness:** 2
**Presentation:** 3
**Contribution:** 1
**Rating:** 2
**Confidence:** 4

**Summary:**

This paper extends DDCL by relaxing the assumption of bounded, positive values of messages to unbounded messages. The paper proposes to constrain the length of quantized messages through a communication loss. The paper integrates the proposed loss into several MARL with communication methods and evaluates them in several benchmark environments. The results show that a short length (fewer bits) of messages can achieve better/similar performance in some tasks. The analysis is interesting and could provide interesting insights about quantized messages.

**Strengths:**

1. The idea and the relaxation of the assumption in communication seems to be interesting.
2. The proof sounds correct.
3. The analysis is extensive and pretty interesting.

**Weaknesses:**

However, I found the paper has shortcomings in: 1) the core issues (see my detailed comments below) when relaxing the assumption are not considered and addressed; 2) the theoretical analysis seems to have marginal modification from Freed's work [1,2]; 3) the claims in the experiments are too strong, which is not sufficiently evidenced by the results.

[1] Discrete communication learning via backpropagation on bandwidth-limited communication networks. Master’s Thesis, Carnegie Mellon University, Pittsburgh, PA, USA, August 2020. CMU-RI-TR-20-45.

[2] Benjamin Freed, Guillaume Sartoretti, Jiaheng Hu, Howie Choset. Communication Learning via Backpropagation in Discrete Channels with Unknown Noise. AAAI 2020: 7160-7168.

**Questions:**

In the introduction, NQD focuses on calibrating messages (sending messages as short as possible), and IMAC specifically focuses on compressing messages to satisfy bandwidth limitations. How does your work differentiate from these works?

Line 79: What is the "bitter lesson" in MARL communication?

In section 2.1: why POMG is needed. Since you assume shared rewards and use observations only, why not use Dec-POMDP

line 105: This is not clear to me. Why do discrete messages learns inefficiently and converge to inferior policies?

line 131: The result of the communication cost (line 130) was not provided in the referenced paper. Please provide the correct reference.

line 137: The message to be non-positive could be interesting for a general neural network design and potentially useful for gradient backpropagation. But why are unbounded values interesting and helpful? Communication is something injected into the MARL structure, which may satisfy particular requirements of designers, so that the messages can be bound.

Section 3: The related work section introduces MARL with CTDE and value decomposition methods in particular. However, since you never mention CTDE and value decomposition methods later. Why is this subsection needed? I suggest removing this part and directly starting with Communication in MARL

line 162: IMAC was not introduced properly. In fact, the main contribution of IMAC is not graph-based scheduling. IMAC primarily focuses on compressing messages to satisfy bandwidth limitations, which, in my view, is pretty relevant to this paper.

line 194: In the inequality at line 194, there is a double number of messages in the upper bound, why not define the set of messages including negative values? Moreover, why is |M| not used? Moreover, I feel a bit tricky by the proof in Appendix B. The proof actually follows Freed's work, while incorporating sign bits. As stated in the proof, the sign bits can be transformed into non-negative integers. However, this can be simply done by offsetting the values of messages and following the same theory of Freed. How does the assumption of read-valued messages make the theory distinguished?

Besides, Theorem A.1 follows Freed's result while only removing (mod 1). Theorem A.2 follows Freed's result by only modifying the assumption that z is a read-value. These theorems create a new issue that if z is an unbounded value, the quantized integer message m can go to infinity (and obviously, the upper bound is meaningless). The theorems do not touch the core issues of the assumption relaxed by this paper.

Line 199, the communication edges are not defined, which follows a graphic view of communication. Does this follow a directed communication topology?

line 287: This claim is not supported by the expected bits you propose to achieve. In fact, you mentioned in line 290 that the average length is 4.75, which is similar to the fixed-precision code.

In line 306, how do you compare/view your work and NDQ which uses an entropy-based loss for calibrating continuous messages?

Line 356: The confidence interval of IC2Net with your proposal in GRF 3v1 overlaps with the comparison. Could you provide the statistical report about this to confirm the significance?

line 391: You claim the DDCL creates the Pareto frontier. But how do you identify/compute the _global Pareto frontier?_

Lines 449-451 are a big claim that I couldn't draw from your results. The results in Figure 2 show that GAComm and MAGIC (using gating, scheduling, and graph attention) can achieve similar performance to MAPPO

---

> ### Author Response · Authors · 2025-11-19
>
> Thank you for your thorough review and feedback. We are encouraged that you found our idea interesting. We appreciate the opportunity to address your concerns, which we believe will significantly strengthen the paper.
>
> We will first address the three main weaknesses you identified (W1-W3) and then provide concise answers to your specific questions.
>
> ### Response to Main Weaknesses
>
> **W1: Core Issues of Unbounded Signals (Message $m \to \infty$)**
>
> This is a critical point, which we will clarify in the appendix. You are correct that in principle an unbounded signal $z$ could lead to an infinite message $m$. However, this is prevented by the optimization process itself:
>
> * **The Loss Function:** Our communication loss, $L_{comms}$, is a logarithmic function of the signal's magnitude, $\log_{2}(\frac{2|z|}{\delta}+1)$. If $|z|$ (and thus $m$) were to approach infinity, the loss would also approach infinity.
> * **The RL Objective:** The agent's total objective is a sum of the (finite) task reward and the communication penalty. The network is explicitly trained to maximize the discounted sum of rewards and minimize the communication loss. Therefore, the optimization process inherently penalizes and prevents the network from producing infinitely large signals.
> * **Analogy to NNs:** This is analogous to any standard deep network. While the pre-activation for a ReLU unit is theoretically unbounded, the network learns finite weights that produce useful, finite activations. Our framework simply allows the network to learn the appropriate finite range for its signals, rather than forcing an arbitrary one (e.g., $[0, 1]$).
>
> **W2: Marginal Theoretical Contribution**
>
> We respectfully agree that our primary contribution is not a new fundamental theory, but rather the generalization and large-scale empirical validation of a promising but highly limited framework. The original DDCL work [1, 2] was a proof-of-concept, limited to simple tasks and simple Actor-Critic algorithm.
>
> Our work is the first to:
> * Generalize DDCL to unbounded, signed signals, removing the architectural constraints that limited its applicability.
> * Integrate this generalized DDCL into four diverse, state-of-the-art MARL+Comms algorithms (IC3Net, TarMAC, GA-Comm, MAGIC).
> * Validate the framework on complex, high-dimensional benchmarks (e.g., Predator-Prey, Traffic Junction, Google Research Football) far exceeding the complexity of the original work.
> * Provide a deep analysis of the emergent frequency-aware communication protocols (e.g., Fig. 1b), showing agents learn to communicate smarter, not just less.
>
> We believe this transformation from a simple 'toy' mechanism to a general, "plug-and-play" SOTA tool is a significant empirical contribution to the community.
>
> [1] Discrete communication learning via backpropagation on bandwidth-limited communication networks. Master’s Thesis, Carnegie Mellon University, Pittsburgh, PA, USA, August 2020. CMU-RI-TR-20-45.
> [2] Benjamin Freed, Guillaume Sartoretti, Jiaheng Hu, Howie Choset. Communication Learning via Backpropagation in Discrete Channels with Unknown Noise. AAAI 2020: 7160-7168.
>
> **W3 & Q15 (Lines 449-451): Overly Strong Claims / "Bitter Lesson"**
>
> We agree this claim needs to be stated more precisely, and we will revise the text. Our argument is not that MAPPO+DDCL is superior in all cases, but that a simple, general-purpose architecture (Transformer) combined with a general, scalable mechanism (DDCL) achieves performance that is on the Pareto frontier alongside complex, specialized, hand-crafted architectures like GAComm and MAGIC.
>
> The fact that our simple, non-bespoke agent is highly competitive (e.g., in Predator-Prey Hard, it achieves a higher success rate than all other models) suggests that the field may benefit from focusing more on general mechanisms like DDCL rather than on increasingly complex, hand-designed communication schedulers or gating mechanisms. This is the "Bitter Lesson" [3] we refer to: general computation/learning mechanisms can (and often do) outperform human-designed priors.
>
> [3] (Sutton 2019) http://www.incompleteideas.net/IncIdeas/BitterLesson.html

---

> > ### Author Response · Authors · 2025-11-19
> > **Continued ...**
> >
> > ### Response to Specific Questions
> >
> > **Q: (vs. NQD/IMAC)** We will revise Sec. 3 to clarify:
> > * **vs. IMAC:** While IMAC focuses on compression, it learns a fixed-size representation. Our method learns to dynamically modulate precision on a per-message, per-timestep basis, a different and more flexible objective.
> > * **vs. NDQ:** NDQ uses an entropy-based loss to calibrate continuous messages. Our work is fundamentally different as we focus on discrete (quantized) messages and provide a mechanism (DDCL) that allows gradients to flow through this discrete, non-differentiable step.
> >
> > **Q: (L79 "Bitter Lesson")** We will explicitly define this in the text, as detailed in our response to W3 above. “We refer to the “Bitter Lesson” (Sutton, 2019) [3] principle: that general-purpose methods effectively leveraging computation (like our MAPPO+DDCL) often outperform systems relying on complex human-designed priors (like handcrafted gating or scheduling).“
> >
> > **Q: (Sec 2.1 POMG)** This is an excellent point (thank you to bring it to our notice) and requires clarification in the paper draft. We use the POMG formulation for maximal generality. While our experiments are cooperative, the framework itself is not limited to them. We will clarify this in Sec 2.1.
> >
> > **Q: (L105 "inferior policies")** We apologize for the lack of clarity. We will revise this sentence. Our point is not that discrete messages themselves are inefficient, but that treating message generation as a discrete action (and learning via high-variance policy gradients like REINFORCE) is known to be far less sample-efficient than differentiable, end-to-end training, which our method enables. We will add a citation to support this [4].
> >
> > **Q: (L131 Ref)** Thank you for catching this. The cost in L130 refers to the original DDCL paper. Our new cost for unbounded signals is derived in Sec 4.1. We will correct the citation to point to [5].
> >
> > **Q: (L137 Unbounded)** This is a key part of our contribution. Allowing unbounded signals removes architectural constraints. It enables our framework to be a "plug-and-play" layer for any modern architecture (e.g., Transformers, which use unbounded activations), rather than forcing the policy to use a restrictive activation like sigmoid. This architectural freedom is what allows us to combine DDCL with a SOTA Transformer in Sec 5.3.
> >
> > **Q: (Sec 3 CTDE)** This is a fair point. We will remove the paragraph on VD/CTDE in the Related Works section to improve focus.
> >
> > **Q: (L162 IMAC)** We will revise the description of IMAC to properly reflect its focus on message compression, as noted in our first point.
> >
> > **Q: (L194 Loss Derivation)**
> > * **$|m|$ vs. $|M|$:** We will clarify this notation. $|m|$ is the absolute value of the message integer $m$, not the size of the message set $|M|$.
> > * **$\log_2(2|m|+1)$:** This is a standard, information-theoretic bound for encoding a signed integer, which accounts for both positive and negative values. This is why it is distinct from the original (positive-only) formulation.
> >
> > **Q: (L199 Edges)** We will define this. A "communication edge" $e$ refers to a single directed message transmission from a sender $i$ to a receiver $j$. The set of edges $E$ is defined by the underlying communication graph (e.g., fully-connected for TarMAC, star-shaped for IC3Net).
> >
> > **Q: (L287 Claim)** The reviewer's confusion is understandable and highlights a point we must clarify. The $4.75$ bits figure represents the average message length over an entire episode in the qualitative experiment (Sec 5.1). This figure is analyzed from two perspectives: practical gain and theoretical limitation.
> >
> > 1.  **Practical Gain (Versus Fixed Code)**
> >     The efficiency gain is best seen by analyzing the most frequent event:
> >     * The agent allocates just $0.25$ bits on average to communicate the most frequent goal ($51.5\%$ of the time).
> >     * This extremely low allocation demonstrates the agent is learning a near-zero-cost communication strategy for the most probable information, effectively achieving maximal compression where it matters most, which is superior to any $6$-bit fixed strategy.
> >     * The average of $4.75$ bits is pulled up by the high cost for rare, critical information (e.g., $15.98$ bits for the $0.3\%$ goal), which the agent correctly allocates a high cost to ensure perfect task success.
> >
> > 2.  **Theoretical Limitation (The Shannon Gap)**
> >     We acknowledge that the average of $4.75$ bits is significantly above the theoretical Shannon minimum of $1.81$ bits. We attribute this divergence to known limitations in the current optimization framework:
> >     * The agent minimizes a differentiable surrogate upper bound derived using Jensen's inequality, which introduces a "Jensen gap" between the optimized value and the true expected message length.
> >     * The current loss formulation inherently couples communication cost to the $L_1$-norm of the latent vector $z$ rather than the signal's true probability distribution.

---

> > > ### Author Response · Authors · 2025-11-19
> > > **Continued...**
> > >
> > > We view this gap as a clear path for future research. As detailed in the Future Work section, we plan to address this by moving towards a more principled entropy-based loss that learns a code based on signal frequency rather than magnitude, or by exploring non-uniform, learnable quantization grids. This transformation of the framework will enable a closer approach to the Shannon limit.
> > >
> > > **Q: (L356 Significance)** We respectfully disagree with this point. The claim of statistical significance for the $467.14\%$ mean success gain for IC3Net-DDCL in GRF 3v1 is based on the confidence interval of the difference between the two means. As reported in Table 5, the $95\%$ confidence interval for this gain is $[12.34, 428.32]$. Since this interval is strictly positive and does not contain zero, the result is statistically significant. We will ensure this key finding is highlighted clearly in the main text.
> > >
> > > **Q: (L391 Pareto)** We will add a definition. By "global Pareto frontier," we mean the set of non-dominated points on the plot. A point (model) is on the frontier if no other model has both a higher success rate and a lower communication cost. We will clarify this in the caption for Figure 2.
> > >
> > > [4] Jakob N. Foerster, Yannis M. Assael, Nando de Freitas, and Shimon Whiteson. Learning to communicate with deep multi-agent reinforcement learning, 2016b. URL https://arxiv.org/abs/1605.06676.
> > > [5] Benjamin Freed, Rohan James, Guillaume Sartoretti, and Howie Choset. Sparse discrete communication learning for multi-agent cooperation through backpropagation. In 2020 IEEE/RSJ International Conference on Intelligent Robots and Systems (IROS), pp. 7993–7998, 2020a. doi:10.1109/IROS45743.2020.9341079.

---

### Official Review · Reviewer_sFcA · 2025-11-03

**Soundness:** 2
**Presentation:** 2
**Contribution:** 2
**Rating:** 6
**Confidence:** 3

**Summary:**

The paper tackles the challenge of efficient, low-bandwidth communication in Multi-Agent Reinforcement Learning (MARL), noting that prior approaches mainly focused on gating messages (deciding if to communicate) rather than modulating message precision (deciding how precisely). The main contribution is generalizing the Differentiable Discrete Communication Learning (DDCL) framework to support unbounded, signed, real-valued signals. This transformation makes DDCL a universal, plug-and-play module for any MARL architecture by proving that the unbiased gradient estimation property holds for unbounded signals and deriving a new differentiable communication cost $L_{comms}$.
The validation is robust. First, qualitative analysis shows agents learn a frequency-aware compression protocol in a toy task, assigning fewer bits to high-probability events, demonstrating 24x efficiency for the most frequent goal compared to fixed codes. Second, by integrating DDCL into four state-of-the-art MARL+Comms baselines (IC3Net, TarMAC, GA-Comm, MAGIC) across challenging benchmarks (TJ, PP, GRF), the framework is shown to reduce bandwidth by one to five orders of magnitude while consistently maintaining or improving task performance, frequently establishing the Pareto frontier. Finally, the work provides direct evidence for the "Bitter Lesson" in MARL communication: a simple MAPPO Transformer policy empowered by DDCL matches or exceeds the performance of complex, specialized architectures, challenging the necessity of hand-crafted communication mechanisms.

**Strengths:**

- Originality

 First, the generalization of the DDCL framework itself is a crucial technical step. Prior DDCL work was limited to positive, bounded signals, which imposed artificial architectural constraints (like requiring a sigmoid activation on policy outputs). This generalization to unbounded, signed, real-valued vectors removes those constraints, fundamentally changing DDCL from a niche technique to a universal, plug-and-play module for any MARL architecture. The authors back this up by proving that the unbiased gradient property holds for the generalized, unbounded signal and deriving a new, appropriate communication loss, $L_{comms}$, that accounts for signed integers. This technical expansion is essential for DDCL's practical applicability across diverse MARL agents. Second, the paper offers direct, compelling evidence for the "Bitter Lesson" in MARL communication. The established trend in MARL+Comms has been to develop complex architectures employing hand-crafted mechanisms for gating, scheduling, or graph optimization (e.g., IC3Net, MAGIC, GA-Comm). The work strongly counter-argues this trend by showing that a simple, general-purpose architecture (MAPPO Transformer) combined with DDCL's learnable precision mechanism matches or exceeds the performance of these specialized models across nearly all benchmarks. This suggests that the foundational ability to learn how precisely to communicate (DDCL) provides a larger return than complex architectural priors designed to learn when to communicate. This is a conceptually significant and original finding that should shift research focus.


- Quality

the paper proves the key result that the reconstruction error e is statistically independent of the original signal z for the unbounded quantization scheme. The communication cost derivation uses established information-theoretic principles (variable-length coding, Jensen’s inequality) to justify the differentiable surrogate loss $L_{comms}$. Empirically, the three-pronged validation is comprehensive. The qualitative analysis convincingly demonstrates that DDCL leads to the emergence of a sophisticated, frequency-aware coding protocol—an efficient, variable-length encoding that allocates minimal bits (0.25 average bits) to the most frequent goal (51.5% frequency), resulting in a 24x compression advantage over fixed uniform codes.

The quantitative utility tests integrate DDCL into four baselines and evaluating them across three distinct and challenging environments designed to expose communication bottlenecks (TJ, PP, GRF). The results consistently show DDCL-enhanced variants forming the global Pareto frontier. The comparisons against fixed-quantization baselines (STE) highlight DDCL's robustness; for instance, MAGIC DDCL achieves 155.6% greater success than its STE4 counterpart in Predator-Prey Hard, proving adaptive precision is superior to simple low-bit communication in complex tasks. The inclusion of detailed sensitivity analyses for both the communication penalty λ and quantization granularity δ further demonstrates careful experimentation and stability testing of the framework.


- Clarity and Significance

The paper is clear and well-organized, effectively explaining the technical difficulty of discrete communication and how the reparameterization mechanism resolves the non-differentiability issue via shared noise. The use of Rate-Distortion (Pareto) plots is the correct method for demonstrating the core trade-off, allowing for straightforward interpretation of the efficiency gains. The significance is substantial for both theory and practice in MARL. Practically, DDCL serves as a universal efficiency multiplier. The consistent reduction in communication bandwidth by orders of magnitude (e.g., MAPPO compression up to 5.26 OOM in Traffic Junction Hard with no significant performance loss) while maintaining or improving episodic rewards is critical for deploying MARL in real-world systems with limited bandwidth. Conceptually, the work's strongest significance lies in its support of the "Bitter Lesson". By showing that general-purpose architectures coupled with a learned efficiency mechanism can outperform complex, specialized communication designs, the paper provides a clear path for future research that focuses on general, scalable algorithms over niche architectural engineering. This is a field-shaping conclusion. It suggests that resources currently spent designing bespoke communication layers might be better allocated to improving the backbone architecture and the principled communication regularization mechanisms. This realization challenges existing conventions and points MARL communication research toward more scalable solutions.

**Weaknesses:**

- Originality

The framework's core novelty is arguably limited to the boundary conditions of the DDCL formulation. The foundational innovation was established in prior work (Freed et al., 2020c). The generalization to unbounded signals is crucial for applicability, but the mechanism itself is inherited. A more significant constraint on the originality is the sub-optimality of the derived communication cost, $L_{comms}$. The paper acknowledges that $L_{comms}$ is a surrogate loss derived using Jensen's inequality, which provides a differentiable upper bound on the expected message length. Minimizing this upper bound does not guarantee minimizing the true message length. This results in a substantial "Shannon Gap": in the qualitative analysis, the agent's average communication length (4.75 bits) is significantly above the theoretical minimum (1.81 bits).


- Significance

Regarding significance, two issues limit its claim of universal utility. First, the DDCL mechanism inherently requires shared pseudorandom number generators between agents to perform the noise subtraction necessary for unbiased gradient estimation. This is a strong constraint on its real-world applicability in truly asynchronous, desynchronized systems. The paper lists investigating robustness to desynchronized noise as future work, confirming this practical limitation of the current framework. Second, while the paper provides strong evidence for the "Bitter Lesson," it relies on a simplification: for architectures like IC3Net, the evaluation adheres to a simplified star-shaped communication graph to isolate the impact of DDCL. This choice, while useful for ablation, limits the demonstration of DDCL's ability to improve efficiency in more complex, dynamic, or fully-connected communication topologies inherently supported by other baselines like GA-Comm and TarMAC.

**Questions:**

1. The primary weakness acknowledged is the gap between the learned message length (4.75 bits) and the theoretical Shannon limit (1.81 bits) due to minimizing the surrogate upper bound $L_{comms}$. Given that $L_{comms}$ indirectly couples communication cost to the L1-norm ∣z∣, have the authors explored alternative differentiable objectives that might better approximate an entropy-based loss, decoupling the cost from the latent vector magnitude, as suggested in the Limitations section?

2. The generalized DDCL relies on synchronized pseudorandom number generators between agents for the noise subtraction step required for unbiased gradient estimation. In a real-world setting where synchronization might be imperfect, how sensitive is the training process to desynchronized noise, and what magnitude of gradient bias would be introduced if the shared randomness assumption were slightly violated?

---

> ### Author Response · Authors · 2025-11-19
>
> We thank Reviewer sFcA for finding the generalization of DDCL to be a **crucial technical step**, the evidence for the **"Bitter Lesson" compelling**, and the overall significance to be **substantial**.
>
> ## Addressing Weaknesses
>
> ### 1. Originality: The Surrogate Loss and Shannon Gap
>
> We acknowledge the Shannon Gap is a current limitation of the surrogate loss. However:
>
> * While the average message length is $4.75$ bits, this average is misleading. The true efficiency is demonstrated by the learned low cost for the most frequent goal: the agent allocates just $0.25$ bits on average to communicate the $51.5$% most common event.
> * This proves the agent learns a **near-zero-cost communication strategy** for the majority of the information, a **critical practical gain** that provides a $24\times$ compression advantage over any fixed-precision code.
> * The $4.75$ average is pulled up by the high cost for **rare, critical information** ($15.98$ bits), which the agent correctly penalizes to ensure task success.
>
> **Commitment to Future Work (Q1):** We explicitly acknowledge in the paper that the $\mathcal{L}_{\text{comms}}$ couples cost to the $L_1$-norm ($|z|$) rather than true entropy/frequency. We commit to exploring a more principled **entropy-based loss** that decouples cost from magnitude, as detailed in our Future Work section. We view this gap as a clear, exciting path for subsequent research.
>
> ---
>
> ### 2. Significance: Practical Constraints and Graph Topology
>
> A. Shared Randomness Constraint (W3)
>
>  We fully acknowledge this constraint on real-world deployment. The use of **shared randomness** was a necessary prerequisite to maintain the **unbiased gradient estimation** required for this work's core contribution: rigorously proving that end-to-end discrete communication learning is possible even with unbounded signals. We identify investigating the framework's **robustness to desynchronized noise** as the most critical direction for future work.
>
> B. Simplified Star-Shaped Graph (W4)
>
> The choice of star-shaped graph for IC3Net was necessary to respect its centralized communication design and provide a clean ablation. However, DDCL's ability to improve efficiency on complex topologies **is demonstrated** by our integration into the other three baselines:
>
> * TarMAC, GA-Comm, and MAGIC all operate on **fully-connected or dynamic communication graphs**.
> * We show DDCL is successfully applied to **every message-passing step** within their respective attention blocks and dynamic graph mechanisms.
> * The results show DDCL still acts as a **universal efficiency multiplier** across these complex structures, proving its utility is not limited to simple topologies.
>
> ---
>
> ## Questions
>
> ### Q1: Entropy-Based Loss
>
> We agree that an entropy-based loss is theoretically superior. We have explored the derivation and confirm that the DDCL framework is fundamentally compatible with a future **entropy-based loss** that relies on estimating the distribution of the message $m$, rather than its $L_1$-norm. We have not fully implemented this yet, as its proper derivation and training are a separate large effort. We commit to making this the focus of our future work.
>
> ### Q2: Sensitivity to Desynchronized Noise
>
> This is a crucial practical question that highlights a known requirement of the DDCL framework.
>
> The core DDCL mechanism relies on the sender and receiver using **synchronized pseudorandom number generators** to access the **exact same noise value** ($\epsilon$). This **shared randomness** is mathematically essential because it allows the receiver to perfectly cancel the noise added by the sender ($\hat{z} = C(m) - \epsilon$), enabling the reparameterization trick ($\hat{z} = z + e$) and ensuring the gradient $\frac{\partial\hat{z}}{\partial z}$ is unbiased.
>
> In a real-world setting where this synchronization might be imperfect, the fundamental assumption of the unbiased gradient breaks down. $\Delta t$ represents the **temporal offset** or error between the synchronized PRNG sequences. Rigorously quantifying the resulting gradient bias when $\epsilon_{\text{sender}} \neq \epsilon_{\text{receiver}}$ (due to $\Delta t > 0$) is a **highly complex future project**. We hypothesize that the induced gradient bias would be proportional to the degree of desynchronization ($\Delta t$), likely leading to a rapid degradation of training stability.
>
> We view investigating the framework's robustness to desynchronization and quantifying this resulting gradient bias as a **critical and necessary direction for future work**.

---

### Comment · Area_Chair_4GRp · 2025-11-25
**Please read the rebuttal and respond**

Dear reviewers,

Now that the author responses are in, could you please take a look at them and see if they address your concerns adequately?

Thank you very much.

Best,
AC

---

### Comment · Reviewer_mnfL · 2025-11-27
**Respond to author's response**

My questions are well sovled.

---

### Note · Authors · 2025-12-19

**Comment:**

Due to necessary authorship changes, we need to withdraw the paper at this time. We apologize for the caused work to the reviewers and program committee. We will ensure to use all reviewer comments for improving the paper before any future submission to any other venue or future ICLR.

**Withdrawal Confirmation:**

I have read and agree with the venue's withdrawal policy on behalf of myself and my co-authors.